# Learning Encoding – Decoding Direction Pairs
# to Unveil Concepts of Influence in Deep Vision Networks

## Abstract

Latent space directions have played a key role in understanding, debugging, and improving deep learning models, since concepts are encoded in directions of the feature space as superpositions. The encoding direction of a concept maps a latent factor to a feature component, while the decoding direction retrieves it. These encoding-decoding direction pairs unlock significant potential to open the black-box nature of deep networks. Decoding directions help attribute meaning to latent codes, while encoding directions help assess the influence of the concept on the predictions, and both directions may assist in unlearning irrelevant concepts. Compared to previous autoencoder and dictionary learning approaches, we offer a different perspective in learning these direction pairs. We base the decoding direction on unsupervised interpretable basis learning and introduce signal vectors to estimate encoding directions. Meanwhile, we empirically prove that the uncertainty region of the model is informative and can be used to effectively reveal meaningful and influential concepts that impact model predictions. Tests on synthetic data show the approach's efficacy in recovering the underlying encoding-decoding direction pairs in a controlled setting, while experiments on state-of-the art deep image classifiers show notable improvements, or competitive performance, in interpretability and influence, compared to previous unsupervised and even supervised direction learning approaches.

## 1. Introduction

Empirical studies indicate that concepts are encoded in feature space directions of deep neural networks (Szegedy et al.,

[1]Anonymous Institution, Anonymous City, Anonymous Region, Anonymous Country. Correspondence to: Anonymous Author <anon.email@domain.com>.

Preliminary work. Under review by the International Conference on Machine Learning (ICML). Do not distribute.

2014; Alain & Bengio, 2018; Zhou et al., 2018; Kim et al., 2018; Elhage et al., 2022; Nanda et al., 2023). The latent factor of a concept constitutes a scalar, signifying the presence of the concept within an image patch. When this scalar is multiplied by the concept's embedding, also referred to as the *encoding* or *signal* direction, this direction maps this factor to a component in the patch's representation. In contrast, a *filter* can be used to extract this latent factor from the representation using the inner product, designating it as a *decoding* direction.

The decoding direction of a concept enables understanding representations, attributing meaning to latent codes (Zhou et al., 2018; Kim et al., 2018), while the encoding direction allows for assessing its influence on the network's predictions (Fel et al., 2023b; Pahde et al., 2024), and both directions may be used in compelling the network to unlearn concepts irrelevant to the prediction task (Anders et al., 2022; Pahde et al., 2023; Dreyer et al., 2024). Most previous approaches (Zhou et al., 2018; Kim et al., 2018; Zhang et al., 2021; Fel et al., 2023b; Doumanoglou et al., 2023; Pahde et al., 2024; Doumanoglou et al., 2024) usually focus on identifying either the decoding or the encoding directions in isolation, limiting their applicability to specific appropriate tasks. Moreover, many of them do not explicitly make this distinction and consider using the concepts' decoding directions in use cases where the encoding direction is a better fit. This has recently been pinpointed in the context of concept influence assessment and model correction in (Pahde et al., 2024).

In this work, we learn the concept encoding-decoding direction pairs, jointly, in an unsupervised manner. Unlike recent advances in sparse autoencoders and dictionary learning (Bricken et al., 2023; Lim et al., 2024; Cunningham et al., 2024), which focus on sparsity within feature space units, we emphasize sparsity in the semantic space of concepts. We model the decoding directions using the principles of the recently introduced, unsupervised interpretable basis learning (Doumanoglou et al., 2023). This modeling provides an explicit rule for concept detection by learning the decoding direction together with an additional threshold to ascertain the presence of a concept. We term this rule as a *concept detector* due to its ability to detect the presence of a concept.

Furthermore, we introduce *signal vectors* as estimators of a concept's encoding direction, confirming their precision in synthetic settings and their impact in real world contexts. Finally, we show that the uncertainty region of a network, that is, the subspace where the network's predictions are uncertain, is informative, and when aligned with the uncertainty region of the concept detectors, can significantly guide the search towards more meaningful concepts that notably impact the network's predictions. Experiments in a controlled setting show the efficacy of the proposed approach in identifying the correct concept encoding-decoding direction pairs, when prior work fails, while experiments on deep vision networks demonstrate the superiority of our method, or competitive performance on par with previous unsupervised and supervised direction learning approaches, in interpretability and influence metrics.

## 2. Related Work

We categorize related work of direction learning into supervised and unsupervised approaches. In each category, we go through previous methods for learning the concept directions, describing the limitations, differences, and similarities with the approach proposed here.

**Supervised Concept Direction Learning** Typical approaches (Zhou et al., 2018; Kim et al., 2018) to interpretable (concept) direction learning use a linear classifier with annotations of a concept dataset. This classifier distinguishes representations of samples with the concept from those without, with interpretable directions as the filter weights and the learned bias as a classification threshold. Known as Concept Activation Vectors (CAVs), these directions resemble the concept decoding directions of Section 1, although they are not exact due to distractor-noise in feature components (Haufe et al., 2014; Kindermans et al., 2017; Pahde et al., 2024). While CAVs could be considered as estimators of concept decoding directions, the recently introduced Pattern-CAVs (Pahde et al., 2024), estimate a concept's encoding direction by the difference in (positive and negative) cluster means.

**Unsupervised Concept Direction Learning** Matrix decomposition methods help identify concept encoding directions without annotations, yet with some limitations. For instance, Principal Component Analysis (PCA) (Graziani et al., 2023) is limited by orthogonality and cannot represent concepts that do not affect variance (Fel et al., 2023a). Likewise, Non-negative Matrix Factorization (NMF) (Zhang et al., 2021; Fel et al., 2023b) assumes positive components and lacks bias, limiting expressivity. While PCA's transpose matrix estimates decoding directions, NMF lacks a simple equivalent. Besides that, for NMF the concept classification rule requires an optimization problem to be solved for every test sample, making the approach computationally more expensive than a calculation of an inner product. Our method overcomes these limitations.

In Dictionary Learning (Bricken et al., 2023; Yun et al., 2023) and Sparse Autoencoders (Sharkey et al., 2022; Cunningham et al., 2024), the goal is to learn decoding-encoding directions by reconstructing representations after decomposing them into latent factors and enforcing sparsity in units of latent variables. Moreover, those factors are constrained to be non-negative. Unlike these methods, our approach allows negative latent factors since it enforces sparsity in the semantic space (a soft-binary vector space) and additionally considers the use of the directions by the model. Beyond that, our method uses a different principle for identifying direction pairs, independent of feature reconstruction.

In contrast to earlier techniques, the method in (Doumanoglou et al., 2023; 2024) learns filter directions of linear classifiers, similar to the supervised methods in (Zhou et al., 2018; Kim et al., 2018). These classifiers convert representations into a soft-binary concept space, guided by a sparsity objective. We ground our approach on this model and additionally enhance it by removing orthogonality constraints, feature space standardization, and adding loss terms to a) sustain or improve interpretability of the identified concepts and b) reduce impact of distractor-noise on filter weights. Although (Doumanoglou et al., 2024) proposed a technique to exploit the utilization of the directions by the network in direction search, our subspace alignment approach shows a notable relative improvement over this previous approach (up to 22.56% in the interpretability metrics), in 3 of 4 cases. Finally, these methods did not consider estimators for the concepts' encoding directions as we do. More details on this comparison can be found in Section H.

## 3. Background

The latent factor of a concept is a scalar linked to the concept's presence, embedded in the feature space via multiplication with its encoding direction, also called the concept's **signal** direction. For this reason, we also refer to this latent factor as the **signal value**. Features are considered as superpositions of signals and noisy directional components called **distractors**. In the proposed approach, a **filter** is a decoding direction that, through the inner product with a feature representation, extracts the signal's value. Below we provide a more formal explanation of these terms and provide details essential to understand our contributions.

### 3.1. Preliminaries

Let $\boldsymbol{X} \in \mathbb{R}^{H \times W \times D}$ denote the representation of an image in an intermediate layer of a convolutional neural network with spatial dimensions $H, W \in \mathbb{N}^+$ and feature space dimensionality $D \in \mathbb{N}^+$. Let also $\boldsymbol{x_p} \in \mathbb{R}^D$ denote an

element of this representation at the spatial location $\boldsymbol{p} = (w, h)$, $w \in \{0, 1, ..., W-1\}$, $h \in \{0, 1, ..., H-1\}$.

### 3.2. Signals, Distractors, Filters, Concept-Detectors, Pattern-CAVs

In encoding a single concept $i$, (Kindermans et al., 2017; Pahde et al., 2024) propose a binary model for the data generation process of feature representations: $\boldsymbol{x_p} = \alpha_{\boldsymbol{p}} \boldsymbol{s}_i + \beta_{\boldsymbol{p}} \boldsymbol{d}$, $\boldsymbol{s}_i, \boldsymbol{d} \in \mathbb{R}^D, \alpha_{\boldsymbol{p}}, \beta_{\boldsymbol{p}} \in \mathbb{R}$. Here, $\boldsymbol{s}_i$ is the signal direction indicating if $\boldsymbol{x_p}$ is part of concept $i$. The key information is in the signal value $\alpha_{\boldsymbol{p}}$. Larger $\alpha_{\boldsymbol{p}}$ suggests greater confidence that $\boldsymbol{x_p}$ belongs to concept $i$. $\boldsymbol{d}$ is the distractor direction, modeling noise, or information unrelated to the concept. $\beta_{\boldsymbol{p}}$ follows a Gaussian distribution $\mathcal{N}(\mu, \sigma^2)$ independent of whether $\boldsymbol{x_p}$ belongs to concept $i$. As per (Kindermans et al., 2017), the value of the signal $\alpha_{\boldsymbol{p}}$ can be extracted via a *regression filter* $\boldsymbol{w}_i$: $z_{\boldsymbol{p},i} = \boldsymbol{w}_i^T \boldsymbol{x_p} = \alpha_{\boldsymbol{p}} \boldsymbol{w}_i^T \boldsymbol{s}_i + \beta_{\boldsymbol{p}} \boldsymbol{w}_i^T \boldsymbol{d}$, if we choose $\boldsymbol{w}_i$ : $\boldsymbol{w}_i \perp \boldsymbol{d}$, and $\boldsymbol{w}_i^T \boldsymbol{s}_i = 1$. Since stronger values of $\alpha_{\boldsymbol{p}}$ indicate more confidence in concept presence, when combined with a threshold $b_i \in \mathbb{R}$ which may be learned from data, this regression filter can be turned into a **concept detector**: $y_{\boldsymbol{p},i} = \sigma(z_{\boldsymbol{p},i} - b_i)$, with $\sigma$ denoting the sigmoid function.

With access to the signal's value, (Haufe et al., 2014; Kindermans et al., 2017) offer a formula to estimate the concept's signal direction:

$$\hat{\boldsymbol{s}}_i = \frac{\text{cov}[\boldsymbol{x_p}, z_{\boldsymbol{p},i}]}{\sigma_{z_{\boldsymbol{p},i}}^2} \tag{1}$$

where, $\sigma_{z_{\boldsymbol{p},i}}^2$ denotes the variance of the signal values in the dataset. This signal estimator relies on signal values, but when trying to explain the latent space, these data are not available. We only have access to $\boldsymbol{x_p}$, while $\boldsymbol{s}_i$ and $\boldsymbol{d}$ are latent variables of the underlying process. For this reason, based on (Haufe et al., 2014; Kindermans et al., 2017), (Pahde et al., 2024) proposed Pattern-CAVs as concept signal estimators that don't need signal values but rely on labeled data, i.e. concept's positive and negative samples. Their method is based on (1), approximating the signal value with binary labels, i.e., $z_{\boldsymbol{p},i} \in \{0, 1\}$, and simplifies to the difference between the means of the concept's positive and negative samples.

### 3.3. Unsupervised Interpretable Direction Learning

Recent research (Doumanoglou et al., 2023) introduced an unsupervised method to identify concepts from the structure of the feature space. Motivated by the directional encoding of concepts, the method partitions the latent space into linear regions, each represented by a hyperplane and a normal vector, forming clusters. Features from an unlabeled *concept dataset*, possibly the network's training set, are assigned to these clusters. It learns $\boldsymbol{W}$ and $\boldsymbol{b}$ of a feature-to-cluster

membership function, a mapping to the semantic space, with $\boldsymbol{y_p} = \sigma(\boldsymbol{W}^T \boldsymbol{x_p} - \boldsymbol{b}) \in [0, 1]^I$, $\boldsymbol{W} \in \mathbb{R}^{D \times I}$, $\boldsymbol{b} \in \mathbb{R}^I$, and $I$ as the cluster count. By softly assigning features to a small number of clusters, interpretability is improved. This is grounded in the idea that an image patch generally holds only a few semantic labels from a larger set, reflecting sparsity in the semantic space. Sparsity in the assignments is achieved using two loss terms: the first is *Sparsity Loss* ($\mathcal{L}^s$), and the second is *Maximum Activation Loss* ($\mathcal{L}^{ma}$), which ensures binary cluster membership.:

$$\mathcal{L}^s = \mathbb{E}_{\boldsymbol{p}}[\mathcal{L}_{\boldsymbol{p}}^s], \quad \mathcal{L}^{ma} = -\mathbb{E}_{\boldsymbol{p}}[\boldsymbol{q}_p^T \log_2(\boldsymbol{y_p})],$$
$$\mathcal{L}_{\boldsymbol{p}}^s = \mathcal{H}(\boldsymbol{q_p}), \quad \boldsymbol{q_p} = \frac{\boldsymbol{y_p}}{||\boldsymbol{y_p}||_1} \tag{2}$$

with $\mathcal{H}$ denoting entropy. The columns of $\boldsymbol{W}$ and elements of $\boldsymbol{b}$ (e.g., $\boldsymbol{w}_i, b_i$) form a linear classifier or concept detector $y_{\boldsymbol{p},i} = \sigma(\boldsymbol{w}_i^T \boldsymbol{x_p} - b_i)$. This method also optimizes linear separability by minimizing the inverse of the classification margin $M_i = \frac{1}{||\boldsymbol{w}_i||_2}$ (*Maximum Margin Loss* - $\mathcal{L}^{mm}$) and penalizes clusters with few assignments using the *Inactive Classifier Loss* - $\mathcal{L}^{ic}$ (Doumanoglou et al., 2024) (More details in Section A). Despite the potential misalignment with human intuition, the sparse nature of transformed representations facilitates concept definition or identification.

### 3.4. Direction Labeling

In (Doumanoglou et al., 2023; 2024), classifier filters form an orthogonal feature space basis, with vectors aligned to cluster regions. Although annotations are not needed for basis learning, interpretability evaluation uses Network Dissection (Bau et al., 2017), a method assigning semantic labels to each vector based on classifier performance with annotated concepts. Despite possible biases against unsupervised learning, we adopt and expand this protocol to evaluate the interpretability of our concept detectors.

### 3.5. Concept Influence Testing

Given an image's intermediate representation for class $k$ and a concept's direction $i$ in latent space, RCAV (Pfau et al., 2020) measures concept sensitivity for the class by perturbing the representation towards the concept's direction with strengh $\alpha$, and subsequently comparing the network's output probability for class $k$ before and after the perturbation. A total dataset score in the range $[-1, 1]$ follows, where zero means inconsistent use of the concept by the model, while extremes indicate strong positive or negative concept contributions to predict class $k$. A statistical test compares concept sensitivity against sensitivity towards random directions to ensure significance. We refer to *directions of significant influence* in cases where the directions meets the criteria of this statistical significance test.

*Figure 1.* **Left**: Intermediate variables: $\mathbf{z_p}, \mathbf{y_p}$, **Middle**: The learnable parameters of the method: $\hat{\mathbf{S}}, \mathbf{W}, \mathbf{b}$. **Right**: Loss terms $\mathcal{L}$ with dependencies on their left. Coral indicates loss contributions of this work, while light yellow indicates loss terms from (Doumanoglou et al., 2023; 2024).

## 4. Method

For a specific network layer, our method receives as input the feature representations of images sourced from a *concept dataset*. The aim of our approach is to learn encoding-decoding direction pairs that correspond to meaningful concepts with influence on the network's predictions. The method is unsupervised, and therefore, the identified concepts may not align with human intuition. However, they reflect clear directional clusters in the feature space of the network. Thus, this approach has the potential to reveal erroneous strategies exploited by the model to make predictions (Section I). In an attempt to make these clusters as meaningful as possible, we optimize for a sparsity property of interpretability in the feature-to-cluster assignments.

We extend the binary signal-distractor data model (Section 3.2) to multiple concepts (Section 4.1) and learn concept detectors $\{\mathbf{W}, \mathbf{b}\}$ using the objectives of Section 3.3. In the new data model, we remove the constraints of (Doumanoglou et al., 2023) on filter orthogonality and feature space standardization, allowing flexible representation clustering. These prior constraints though, acted as regularizers to prevent degenerate solutions; thus, we address their removal with additional loss terms discussed in Section 4.2 that sustain or even improve the interpretability of the clustering. We additionally estimate concept signal directions using learnable **signal vectors** $\hat{s}_i$ (Section 4.3). We also propose Uncertainty Region Alignment (Section 4.4), a loss that significantly improves cluster quality. In summary, concept detectors and signal vectors are learned together in an end-to-end process, influenced by the losses of Sections 3.3, 4.2, 4.3 and 4.4. A summary of the interconnections between components of the method is provided in Fig. 1.

### 4.1. Multi-Concept Signal-Distractor Data Model

We introduce an extended signal-distractor model for the latent space, encoding multiple concepts. Each spatial element $\mathbf{x_p}$ is a linear combination of latent concept signals $\mathbf{S} \in \mathbb{R}^{D \times I}$ and distractors $\mathbf{D} \in \mathbb{R}^{D \times F}, F \leq D - I$

$$\mathbf{x_p} = \mathbf{S}\boldsymbol{\alpha_p} + \mathbf{D}\boldsymbol{\beta_p} \qquad (3)$$

with $\boldsymbol{\alpha_p} \in \mathbb{R}^I$ and $\boldsymbol{\beta_p} \in \mathbb{R}^F$. $\mathbf{S}$ is a matrix of $I \in \mathbb{N}^+$, $D$-dimensional, unit-norm concept signal directions and $\mathbf{D}$ a matrix denoting a basis for distractor components. Each signal direction encodes the presence of a distinct concept. We apply the same assumptions for individual signal values $\alpha_{\mathbf{p},i}$ (the $i$-th element of $\boldsymbol{\alpha_p}$) and distractor coefficients $\beta_{\mathbf{p},f}$ as in Section 3.2. Finally, we further assume that only a limited number of semantic concepts are assigned to $\mathbf{x_p}$, among many possible semantic labels.

### 4.2. Interpretability Losses to Recover Implicit Regularizations

We propose **Self-Weighted Reduction** ($\mathcal{R}_{SW}$) as a loss aggregation method to optimize upper bounds. Consider a set of un-reduced loss values $\mathcal{L}_k, k \in \mathbb{N}$. The Self-Weighted Reduction is:

$$\mathcal{R}_{SW}(\{\mathcal{L}_k\}) = \frac{\sum_k \mathcal{L}_k^{\nu+1}}{\sum_k \mathcal{L}_k^{\nu}} \qquad (4)$$

which is equal to the weighted average of elements in $\{\mathcal{L}_k\}$ with each element being weighted by $\mathcal{L}_k^{\nu}, \nu > 1, \nu \in \mathbb{R}^+$ a sharpening factor. This loss may be seen as a soft-differentiable version of the max operation, since the largest value in the set of $\{\mathcal{L}_k\}$, is weighted with the largest weight.

**Excessively Active Classifier Loss** ($\mathcal{L}^{eac}$) This loss penalizes excessively large clusters to prevent trivial solutions where all inputs belong to one cluster. It relies on a hyperparameter $\rho$, similar to sparse autoencoders (Ng et al., 2011), which sets a proportional bound on cluster size. The unreduced formula is below, with $\gamma > 1, \gamma \in \mathbb{R}^+$ as a sharpening factor and $1 - \rho$ normalizing the loss in range $[0, 1]$:

$$\mathcal{L}_i^{eac} = \frac{1}{1 - \rho}\text{ReLU}(\mathbb{E}_{\mathbf{p}}[y_{\mathbf{p},i}^{\gamma}] - \rho) \qquad (5)$$

The final reduced loss, is using $\mathcal{R}_{SW}$: $\mathcal{L}^{eac} = \mathcal{R}_{SW}(\{\mathcal{L}_i^{eac}\})$

**Sparsity Bound Loss** ($\mathcal{L}^{sb}$) With this loss term we minimize the upper bound of the un-reduced $\mathcal{L}^s$, among pixel locations, using $\mathcal{R}_{SW}$. In more detail, if $\mathcal{L}_{\mathbf{p}}^s$ (2) denotes the Sparsity Loss for pixel $\mathbf{p}$, the Sparsity Bound Loss ($\mathcal{L}^{sb}$) is defined as $\mathcal{L}^{sb} = \mathcal{R}_{SW}(\{\mathcal{L}_{\mathbf{p}}^s\})$

### 4.3. Signal Vectors as Concept Signal Estimators

Considering the new data model outlined in Section 4.1, the assumptions of (1) for estimating a concept's signal direction may not be valid. Specifically, there may be an

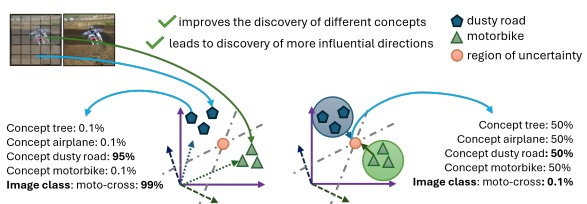

*Figure 2.* The uncertainty region of the network is defined as the subspace where all network's predictions are maximally uncertain. The uncertainty region of the concept detectors is defined as the intersection of all their decision hyperplanes. Aligning these two through feature manipulation improves the interpretability and influence of the identified concepts.

anti-correlated relationship between the variables $\alpha_{p,i}$ and $\alpha_{p,j}, i \neq j$ due to the fact that concept labels are sparsely assigned to each $x_p$, indicating mutual exclusivity in concept label attributions. Nevertheless, as detailed in the Section B, we can effectively apply (1) by only using positive samples of the concept ($p : y_{p,i} > 0.5$) when calculating variance and covariance, rather than both positive and negative samples as previously recommended. We call the signal estimator for concept $i$, obtained under these conditions, the *signal vector* $\hat{s}_i$. However, we still require access to signal values. As explained in Section 3.2, estimating signal values can be attributed to the concept detectors' filters. They can serve as signal value estimators if the weight vector $w_i$ is orthogonal to all $s_j$ where $j \neq i$, as well as the distractor subspace $D$.

Thus, we employ the folllowing Filter-Signal Vector Orthogonality Loss to learn the directions:

$$\mathcal{L}^{fso} = \sqrt{\mathbb{E}_{i,j}\left[(1 - \delta_{i,j})\bar{w}_i^T \bar{s}_j)^2\right]} \qquad (6)$$

with $\delta_{i,j}$ the kronecker delta and $\bar{w}, \bar{s}$ denoting the L2-normalized filter weights and signal vectors. To achieve accurate signal value extraction, $w_i$ should additionally be orthogonal to the distractor basis; however, we do not explicitly estimate the distractors. Instead, we use the Uncertainty Region Alignment loss from Section 4.4 to ensure alignment of the directions with utilization by the network.

## 4.4. Uncertainty Region Alignment to Improve Interpretability and Influence

The presence or absence of a concept in a representation can offer neutral, supportive, or opposing evidence against the prediction of a class. Since the concept-class pair association is unknown when learning concept directions, a straightforward strategy to perform concept arithmetic on the features in order to find their utility by the network lacks ground-truth information on how concepts affect class predictions. To overcome this difficulty, we can make a simple but more elegant hypothesis that uncertain network predic-

tions occur when the representation has ambiguous concept information. We propose improving the direction search by aligning the uncertainty regions of the network and the concept detectors. The *uncertainty region of the network* is the subspace where its predictions are most uncertain, and the *uncertainty region of the concept detectors* is the subspace where their decision hyperplanes intersect. Figure 2 illustrates the concept of Uncertainty Region Alignment.

We first manipulate spatial features $x_p$ towards the direction $-dx_p$ to arrive in $x'_p = x_p - dx_p$. Based on our estimates of $w_i$, $b_i$, and $\hat{s}_i$, we select the direction $dx_p$ so that the shifted $x'_p$ lies at the intersection of the concept detectors' decision hyperplanes. Then, we ensure the network's predictions for these features are highly uncertain, effectively aligning both uncertainty regions.

### Unconstrained and Constrained Uncertainty Region Losses ($\mathcal{L}^{uur}, \mathcal{L}^{cur}$)

We define two types of Uncertainty Region Loss: i) unconstrained $\mathcal{L}^{uur}$ and ii) constrained $\mathcal{L}^{cur}$. Each loss uses a different feature manipulation strategy $dx_p$ but both share the same final formula

$$\mathcal{L}^{uur} = \mathcal{L}^{cur} = -\mathbb{E}_{X'}\left[\mathcal{H}(f^+(X'))\right] \qquad (7)$$

(with $\mathcal{H}$ denoting entropy and $f^+$ denoting the part of the network after the layer of study providing output class probabilities). In (7), $X'$ denotes a manipulated image representation, with every $x_p$ shifted in the direction $-dx_p$.

**i) Unconstrained Uncertainty Region Manipulation** Suppose that we know the concept decoding directions $w_i$ and the classification thresholds $b_i$, we can bring all $x_p$ to the concept detectors' uncertainty region by manipulating each $x_p$ in the direction $-dx_p$ with the following formula:

$$w_i^T x'_p - b_i = 0 \Rightarrow w_i^T(x_p - dx_p) - b_i = 0, \forall i,$$
$$W^T(x_p - dx_p) - b = 0 \Rightarrow dx_p = (W^T)^+(W^T x_p - b)$$

with $A^+$ denoting the pseudo-inverse of $A$.

**ii) Constrained Uncertainty Region Manipulation** The previous unrestricted approach to the manipulation of features might lead to datapoints falling outside the concept encoding manifold of the network, causing an unfaithful alignment. To address this, we suggest restricting feature manipulation to occur within the span of the signal vectors, i.e. $dx_p = \hat{S}v, v \in \mathbb{R}^I$, and thus:

$$W^T(x_p - dx_p) - b = 0 \Rightarrow W^T(x_p - \hat{S}v) - b = 0 \Rightarrow$$
$$W^T\hat{S}v = W^T x_p - b \Rightarrow v = (W^T\hat{S})^+(W^T x_p - b) \Rightarrow$$
$$dx_p = \hat{S}v = \hat{S}(W^T\hat{S})^+(W^T x_p - b)$$

where $\hat{S}$ represents a matrix whose $i$-th column is equal to the estimated signal vector $\hat{s}_i$.

*Table 1.* Evaluating the performance of the concept detectors in classifying pixel representations in the experiment on synthetic data. The metric is Intersection over Union (IoU). Rows correspond to concept detectors and columns to ground-truth concept classes.

| | | Concept | | |
|---|---|---|---|---|
| | | **#0** | **#1** | **#2** |
| **Detector** | **#0** | 0 | 0.96 | 0 |
| | **#1** | 0.92 | 0 | 0 |
| | **#2** | 0 | 0 | 0.96 |

# 5. Experiments

## 5.1. Experiment on Synthetic Data

In this section, we test our method on synthetic data, demonstrating that it reliably identifies the key elements of the data generation process detailed in Section 4.1, under challenging conditions for conventional techniques.

We generate features $x_p$ according to (3) by setting the embedding space dimensionality to $D = 16$, the number of distinct concepts to $I = 3$, the size of the distractor basis to $F = 2$ and randomly create unit-norm vectors to construct the matrices $S$ and $D$. The latent signal values and distractor coefficients follow the uniform distribution: $\alpha_{p,i} \sim \mathcal{U}(0.0, 2.5)$ if $p$ is not part of concept $i$ and $\alpha_{p,i} \sim \mathcal{U}(2.5, 5.0)$, otherwise, while $\beta_{p,f} \sim \mathcal{U}(0, 5.0)$ is independent of the pixel concept label. We introduce a bias of 10 across all dimensions of the representations to maintain them in the positive quartile, similar to the impact of a ReLU layer.

The image representations are considered with two spatial elements $p_1, p_2$, where $W = 2$ and $H = 1$. Each pixel representation corresponds to a single concept. Let $c(p) \in \{0, 1, 2\}$ represent the concept label of $p$, and $k \in \{a, b, c\}$ denote an image class. We construct image representations as follows: for $k = a$, $c(p_1) = 0$ and $c(p_2) = 1$; for $k = b$, $c(p_1) = 0$ and $c(p_2) = 2$; and for $k = c$, $c(p_1) = 1$ and $c(p_2) = 2$. We generate a balanced dataset with each class being represented by 1000 images.

The network we use is composed of just two layers (corresponding to the top part of a potentially larger convolutional network). The first is an average-pooling layer, and the second is a linear layer with $K = 3$ output classes. After training, the network attains 99% accuracy on a test set, randomly generated based on the previous principles. More details of this experiment are in Section E.

**Decoding directions:** We first assess the ability of the concept detectors to identify the specified concepts. Table 1 shows Intersection over Union scores for each detector against actual concept classes. Zero values indicate complete purity and no mixing of the concepts, and all scores are above 0.92, showing success. We also examine how well

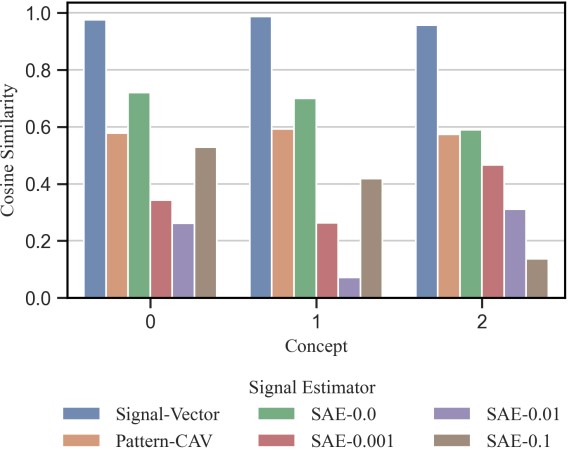

*Figure 3.* Cosine Similarity of ground-truth concept encoding directions and their estimation: Signal Vectors, Pattern-CAVs, Sparse AutoEncoder with regularization $\lambda = \{0.0, 0.001, 0.01, 0.1\}$. The proposed Signal Vectors were the most accurate by a large margin, followed by an Auto Encoder without sparsity constraints ($\lambda = 0$) and Pattern-CAVs.

the learned filters extract signal values from representations. Our method identifies signal values as a deviation from the dataset's average since distractor directions aren't directly estimated (See also Section C). The Root Mean Squared Error (RMSE) between these extracted values and the ground truth, after subtracting the mean signal value, is noted as 0.26.

**Encoding directions:** We also examine the cosine similarity between signal vectors and true concept encoding directions, comparing them with a Sparse Autoencoder (Bricken et al., 2023) at varying sparsity levels and Pattern-CAVs learned using ground-truth labels. Fig. 3 shows that signal vectors closely estimate the concept's encoding directions, unlike Pattern-CAVs, which falter due to the different assumptions made about the data. The sparse autoencoder fails due to the fact that the reconstruction goal objective can be met with any basis of the data manifold.

The ground-truth encoding directions of this example (Table 6) contain both positive and negative components and the relationship among the encoding directions (and the distractors) is in general not orthogonal. In theory and without the need for practical experiments, this example cannot be addressed by NMF, K-Means, or PCA. NMF would produce a signal basis of non-negative components akin to the cluster centers of K-Means, which would point toward the positive quartile where the centroids reside. Moreover, the non-orthogonal nature of the ground truth encoding directions implies that the PCA's solution space is insufficient.

*Table 2.* Ablation study wrt interpretability losses (top part) and uncertainty region alignment losses (bottom part).

| | | | ResNet18 / Places365 | | | | | | |
|---|---|---|---|---|---|---|---|---|---|
| $I$ | $\mathcal{L}^{uur}$ | $\mathcal{L}^{sb}$ | $\mathcal{L}^{eac}$ | $\mathcal{L}^{cur}$ | $\mathcal{L}^{fso}$ | $\mathcal{S}^1$ | $\mathcal{S}^2$ | SDC | SCDP |
| | ✓ | ✗ | ✗ | ✗ | ✗ | **59.06** | 25.91 | **377** | 2487 |
| 500 | ✓ | ✓ | ✗ | ✗ | ✗ | 49.02 | 35.23 | 354 | 2480 |
| | ✓ | ✓ | ✓ | ✗ | ✗ | 54.55 | 37.38 | 359 | 2118 |
| | ✗ | ✓ | ✓ | ✓ | ✓ | 57.34 | **38.36** | 376 | **3271** |
| | ✓ | ✓ | ✓ | ✗ | ✗ | 50.49 | 34.76 | 283 | 1451 |
| 450 | ✗ | ✓ | ✓ | ✓ | ✗ | 50.94 | **36.01** | 335 | 2930 |
| | ✗ | ✓ | ✓ | ✓ | ✓ | **52.63** | 34.86 | **360** | **2956** |

## 5.2. Experiment on Deep Image Classifiers

We evaluate the method's components through practical experiments on the final convolutional layer of a ResNet18 (He et al., 2016) trained on Places365 (Zhou et al., 2017). For comparison with earlier unsupervised approaches, we also experimented with a ResNet50 trained on Moments in Time (Monfort et al., 2019). Unless stated otherwise, our Encoding-Decoding Direction Pairs (EDDP) uses a weighted combination of all losses from Sections 3.3 and 4.2, along with $\mathcal{L}^{fso}$ and $\mathcal{L}^{cur}$ from Sections 4.3 and 4.4. The hyperparameter $I$ is set to $I = 500$, with other method parameters detailed in the Section F.

**Evaluation of the Decoding Directions: Interpretability** Our method's effectiveness in identifying meaningful concepts is assessed using a quantitative approach measuring the interpretability of the directional clustering obtained by the learned concept detectors. We follow the protocol in (Doumanoglou et al., 2023). We utilize the Broden (Bau et al., 2017) dataset for ResNet18, and Broden Action (Ramakrishnan et al., 2019) for ResNet50 to learn and label directions (Section 3.4). The datasets feature dense pixel annotations; Broden includes 1197 concepts across $63K$ images in 5 concept categories (object, part, material, texture, color), while Broden Action incorporates an additional action category with 210 labels and $23K$ more images.

We employ two metrics from (Doumanoglou et al., 2023). Specifically, let $\phi_i(c, \mathcal{K})$ the Intersection Over Union for concept detector $i$ in identifying concept $c$ within the dataset $\mathcal{K}$. Define $c_i^\star = \arg\max_c \phi_i(c, \mathcal{K}_t)$, indicating the concept label detected best by concept detector $i$ within the training subset of the dataset ($\mathcal{K}_t$). With $\mathcal{K}_v$ as the validation subset, our interpretability scores $\mathcal{S}^1$ and $\mathcal{S}^2$ are:

$$\mathcal{S}^1 = \int_0^1 \sum_{i=0}^{I-1} \mathbb{1}_{x \geq \xi}\big(\phi_i(c_i^\star, \mathcal{K}_v)\big) d\xi \qquad (8)$$

$$\mathcal{S}^2 = \int_0^1 |\{c_i^\star \,|\, \exists i : \phi_i(c_i^\star, \mathcal{K}_v) \geq \xi\}| d\xi \qquad (9)$$

The first metric $\mathcal{S}^1$ counts concept detectors with an IoU performance that exceeds a score threshold $\xi$. The second

metric $\mathcal{S}^2$ uses the cardinality of the set $|.|$ to count the unique concept labels detected by the concept detectors with IoU above $\xi$. Both metrics become threshold-agnostic, by integrating on all $\xi \in [0, 1]$. Qualitative segmentations using the learned concept detectors appear in Section G.

**Evaluation of the Encoding Directions: Influence** We assess the ability of our method to identify influential concepts to model predictions using RCAV (Pfau et al., 2020) (Section 3.5). For sensitivity scores, we spatially replicate signal vectors or Pattern-CAVs. Direction significance is tested with RCAV's label permutation test to generate random directions, with the significance threshold set to 0.05 and Bonferroni correction. Two metrics summarize the results: Significant Direction Count (SDC) and Significant Class-Direction Pairs (SCDP). SDC is the count of signal vectors that significantly influence at least one model class, while SCDP tallies class-direction pairs where vectors significantly affect the class. In ablation studies excluding $\mathcal{L}^{cur}$, we report influence metrics for signal vectors estimated **post learning the directions** with the conditions discussed in Section 4.3. Network explanations using the learned encoding directions and RCAV are in Section F.2.

**Ablation Study: Interpretability Losses** The top part of Table 2 presents the outcome of an ablation study focusing on the interpretability loss terms introduced in Section 4.2. We start with only $\mathcal{L}^{uur}$ and progressively incorporate $\mathcal{L}^{sb}$ and $\mathcal{L}^{eac}$, observing a steady and notable improvement in $\mathcal{S}^2$, which is more challenging to optimize compared to $\mathcal{S}^1$, while still keeping up with performance in terms of $\mathcal{S}^1$. Eventually, retaining the interpretability terms and transitioning to the use of the proposed signal vectors and $\mathcal{L}^{cur}$ we see further improvement in interpretability, and a significant increase in the SCDP influence metric.

**Ablation Study: Uncertainty Region Alignment and Signal Vectors** The lower section of Table 2 presents the metric scores of an ablation study centered on uncertainty region alignment and signal vectors. By moving from $\mathcal{L}^{uur}$ to $\mathcal{L}^{cur}$ and subsequently incorporating $\mathcal{L}^{fso}$, we observe a steady enhancement across all influence metrics. The table highlights the substantial effect of employing signal vectors (used in $\mathcal{L}^{cur}$) to detect influential directions. Although the signal vectors derived from the combination of $\mathcal{L}^{cur}$ and $\mathcal{L}^{fso}$ prove to be more influential than the others, this comes at a minor reduction in interpretability. This trade-off between interpretability and influence is aligned with the observations in (Kim et al., 2018; Pfau et al., 2020), where non-interpretable random directions can significantly affect a model's predictions.

**Interpretability Comparison with Unsupervised Approaches** Previous unsupervised approaches (Zhang et al., 2021; Graziani et al., 2023; Fel et al., 2023b) lack a clear classification rule for concept detection, impeding quan-

*Table 3.* Comparison of concept-detectors (CDs)' performance in pixel classification and image segmentation tasks. Comparing between: a) individual CDs, b) combined CDs (*Linear-OR*) c) individual CDs with their thresholds learned with supervision, and d) IBD: a set of classifiers learned in a supervised way.

| | mPrecision | mRecall | mAP | mF1Score | $\mathcal{S}^1$ | $\mathcal{S}^2$ | mIoU |
|---|---|---|---|---|---|---|---|
| IBD (Zhou et al., 2018) | 0.84 | 0.6 | 0.77 | 0.69 | 53.32 | 53.32 | 0.20 |
| EDDP Individual (ours) | 0.81 | 0.24 | 0.53 | 0.33 | 57.33 | 38.35 | 0.11 |
| EDDP Linear-OR (ours) | 0.73 | 0.4 | 0.59 | 0.45 | 30.56 | 30.56 | 0.11 |
| EDDP Individual /w sup thresholds (ours) | 0.62 | 0.49 | 0.52 | 0.53 | N/A | N/A | N/A |

*Table 4.* Influence Comparison against Pattern-CAVs.

| RCAV $\alpha$ | 0.5 | 2.0 | 5.0 |
|---|---|---|---|
| $\mathcal{S}^3$ | 0.37 | 0.37 | 0.38 |

*Table 5.* Interpretability comparison with prior work on unsupervised basis learning. Works considered: UIBE (Doumanoglou et al., 2023), CBE (Doumanoglou et al., 2024) and CBE with CNN Classifier Loss replaced with the proposed $\mathcal{L}^{uur}$ (CBE /w $\mathcal{L}^{uur}$).

| | **ResNet18 / Places365** | | |
|---|---|---|---|
| | UIBE | CBE | CBE /w $\mathcal{L}^{uur}$ |
| $\mathcal{S}^1$ | 60.93 (+0.0%) | **69.43 (+13.95%)** | 67.3 (+10.45%) |
| $\mathcal{S}^2$ | 28.39 (+0.0%) | 31.53 (+11.06%) | **32.16 (+13.28%)** |
| | **ResNet50 / MiT** | | |
| | UIBE | CBE | CBE /w $\mathcal{L}^{uur}$ |
| $\mathcal{S}^1$ | 124.73 (+0.0%) | 131.73 (+5.61%) | **158.76 (+27.28%)** |
| $\mathcal{S}^2$ | 18.47 (+0.0%) | 26.94 (+45.86%) | **33.02 (+78.78%)** |

titative evaluation, and relying primarily on human assessment of interpretability. Our work overcomes these limitations, enabling a more effective quantitative evaluation. We compare in interpretability terms with the work of (Doumanoglou et al., 2023; 2024). Since we base our method on them, the meaningful aspect to compare is the contribution of our $\mathcal{L}^{uur}$ (and not $\mathcal{L}^{cur}$ since those works did not consider concept encoding directions). We use the exact setup of (Doumanoglou et al., 2024) (i.e. without lifting the orthogonality of the directions, the feature standardization, or considering signal vectors) and replace their *CNN Classifier Loss* with our $\mathcal{L}^{uur}$. The experimental results are provided in Table 5. Our Uncertainty Region Alignment loss significantly improves the interpretability metrics in 3 of 4 scenarios (by up to +22.56% relative improvement) while remaining competent in the remaining case. This justifies that our alignment of uncertainty regions is more effective than the technique proposed in (Doumanoglou et al., 2024).

**Interpretability Comparison with a Supervised Approach** We compare the concept classifiers learned using our method against those learned through a supervised approach, with a focus on interpretability (Table 3). We calculate averaged binary classification metrics across detectors. We consider three variants of the proposed method: a) independent learning of directions (the exact outcome of our method), b) combining directions with a shared label (post initial leaning) using a linear layer for classification (this layer classifies representations as positive if any detector with the same label does), and c) considering the learned

directions but optimizing the classification threshold in a supervised manner to enhance the F1 Score. This last approach assesses direction quality independent of bias. Results show that individual classifiers from the proposed method achieve high precision, comparable to supervised ones, but suffer from low recall due to their sparsity-driven objectives. Combining classifiers with the same label improves recall, while supervised optimization of the bias further enhances F1 Scores by reducing sparsity.

**Influence Comparison with a Supervised Approach** We compare network sensitivity to Pattern-CAVs and signal vectors. Let $j \in \{0, 1, ..., N_l - 1\}$ index concept detectors with the same concept label $l$, and $S^l_{j,k}$ represent the RCAV sensitivity score of class $k$ relative to the signal vector of the $j$-th concept detector for label $l$. Similarly, $S^l_{P,k}$ is the sensitivity score of class $k$ relative to the Pattern-CAV for the same label. Pattern-CAVs use ground-truth pixel-level labels. Network Dissection can assign identical labels to multiple detectors, making direct comparisons with Pattern-CAVs challenging. Inspired by RCAV, we regard signal vectors as *noise vectors* and assess Pattern-CAV sensitivity for a label against them. We define a metric $\mathcal{S}^3$ which when above 0.5 indicates Pattern-CAVs have more network influence than signal vectors at significance level $\theta = 0.05$ (Bonferroni correction applies):

$$\mathcal{S}^3 = \mathbb{E}_{l,k}\left[\mathbb{1}(p_{l,k} < \frac{\theta}{N_l})\right]$$

$$p_{l,k} = \frac{1}{N_l}\sum_j \mathbb{1}\left(|S^l_{j,k}| \geq |S^l_{P,k}|\right)$$

Metrics for different RCAV values $\alpha$ are in Table 4. The $\mathcal{S}^3$ scores are below 0.5, indicating that Pattern-CAVs are less influential than signal vectors on network predictions.

## 6. Conclusion

We introduced an innovative unsupervised technique to uncover pairs of latent space encoding-decoding directions that align with interpretable and influential concepts. This research offers a new perspective on the unsupervised identification of concept directions, unlike previous methods based on feature reconstruction or decomposition, paving the way for additional exploration.

## Impact Statement

This paper presents work whose goal is to advance the field of Machine Learning. There are many potential societal consequences of our work, none which we feel must be specifically highlighted here.

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

## A. Unsupervised Interpretable Basis Extraction and Concept-Basis Extraction Losses

**Sparsity Loss** ($\mathcal{L}^s$) (Doumanoglou et al., 2023)

Based on the observation that the number of semantic labels that may be attributed to an image's patch, are only a fraction of the set of possible semantic labels, this loss enforces sparsity across the classification results $y_{\boldsymbol{p},i}$ for each spatial representation $\boldsymbol{x_p}$. In particular, the sparsity loss for pixel $\boldsymbol{p}$ is defined as:

$$\mathcal{L}^s_{\boldsymbol{p}} = -\sum_i q_{\boldsymbol{p},i}\log_2 q_{\boldsymbol{p},i}, \quad q_{\boldsymbol{p},i} = \frac{y_{\boldsymbol{p},i}}{\sum_i y_{\boldsymbol{p},i}} \tag{10}$$

and the aggregated sparsity loss $\mathcal{L}^s$:

$$\mathcal{L}^s = \mathbb{E}_{\boldsymbol{p}}\big[\mathcal{L}^s_{\boldsymbol{p}}\big] \tag{11}$$

**Maximum Activation Loss** ($\mathcal{L}^{ma}$) (Doumanoglou et al., 2023)

With the complement of this loss the pixel classifications are enforced to become binary:

$$\mathcal{L}^{ma} = \mathbb{E}_{\boldsymbol{p}}\Big[-\sum_i q_{\boldsymbol{p},i}\log_2 y_{\boldsymbol{p},i}\Big] \tag{12}$$

**Inactive Classifier Loss** ($\mathcal{L}^{ic}$) (Doumanoglou et al., 2024)

This loss ensures that each classifier in the set, classifies positively at least $\nu \in [0,1]$ percent of pixels in the concept dataset.

$$\mathcal{L}^{ic} = \mathbb{E}_i\Big[\frac{1}{\nu}\text{ReLU}\big(\nu - \mathbb{E}_{\boldsymbol{p}}[y^{\gamma}_{\boldsymbol{p},i}]\big)\Big] \tag{13}$$

with $\nu = \frac{\tau}{I}, \gamma > 1, \gamma \in \mathbb{R}^+$ denoting a sharpening factor and $\tau \in [0,1]$ denoting a percent of pixels in the dataset to be evenly distributed among the $I$ classifiers in the set.

**Maximum Margin Loss** ($\mathcal{L}^{mm}$) In the original formulation of (Doumanoglou et al., 2023), the Maximum Margin Loss was defined as $\mathcal{L}^{mm} = \frac{1}{M}$ with $M$ being a single parameter for the whole set of classifiers since the optimization was performed in the standardized space with shared parameters for the margins $M$ and biases $b$. In this work, we removed the standardized space constraints and instead, we have a margin parameter $M_i$ for each classifier in the set. Thus, we modify the Maximum Margin loss to become:

$$\mathcal{L}^{mm} = \frac{1}{I}\sum_i \frac{1}{M_i} \tag{14}$$

## B. Signal Direction Estimation

In the formulation of signal-distractor data-model defined in (Kindermans et al., 2017) and detailed in Section 3.2, (1) is an accurate estimator of a concept's encoding direction whenever the non-signal (here distractor) components in the data contain information independent of whether the representation $\boldsymbol{x_p}$ belongs to concept $i$. The formula exploits the fact that $\text{cov}[z_{\boldsymbol{p}}, \boldsymbol{d}]$ should be 0. An important role in this discussion has the threshold $b_i$ that delimits the positive samples of a concept. When considering the encoding of multiple concepts, like the data model that we proposed in Section 4.1, it is reasonable to make the assumption of (Doumanoglou et al., 2023) that concept label attributions are mutually-exclusive (e.g. when an image patch corresponds to concept *tree* it is not *car* or *sky* or ...). Thus, the signal values $a_{\boldsymbol{p},i}, a_{\boldsymbol{p},j}$ in $\boldsymbol{x_p}$ **may** not be independent but anti-correlated since for a pixel $\boldsymbol{p}$ of concept $i$ $a_{\boldsymbol{p},i} > b_i$ and $a_{\boldsymbol{p},j} < b_j$, possibly violating the assumptions of (1) regarding independence. Whether the violation is significant or not may depend on the relationship between $b_i$ and the mean of $a_{\boldsymbol{p},i}$ as well as whether we consider a balanced dataset. This assumption is not violated though, if we consider only the reference samples that belong to the concept. In that case, among that subset of the data, the signal values $a_{\boldsymbol{p},i}$ and $a_{\boldsymbol{p},j}$ are now independent by assumption, as we now removed the biases $b_i, b_j$ due to sub-sampling. This allows us to still consider (1) as a signal estimator, even under the extended data model of multiple concepts, provided that we subsample the data based on their concept label.

## C. Extracting Signal Values with the Filters of Concept Detectors

Starting from the data model of Section 4.1 for the encoding of multiple concepts in the representation, we have:

$$x_p = S\alpha_p + D\beta_p$$

As discussed in Section 4.3, signal values, which are required to estimate the encoding direction of a concept, are extracted using the filter weights of the concept detectors. Yet, as we discussed in that Section, in order for this to happen, the filter weights $w_i$ need to be orthogonal to $s_j$ and $D$. Since we do not explicitly estimate distractors in this work, there maybe an innevitable error when extracting the value of the signal (we say maybe, because this might also be mitigated by the Uncertainty Region Alignment losses). Here we study on the order of this error.

$$z_p = w_i^T x_p = w_i^T S\alpha_p + w_i^T D\beta_p$$
$$\frac{z_p}{w_i^T s_i} = a_{p,i} + \frac{w_i^T D\beta_p}{w_i^T s_i} \tag{15}$$

Thus, we can estimate the signal value, by the inner product between $w_i$ and $x_p$ and divide by $w_i^T \hat{s}_i$.

In the experiment on Synthetic Data (Section 5.1) we introduced an additional constant bias of 10 to bring all the feature representation in the positive quartile. This changed the above data model to:

$$x_p = S\alpha_p + D\beta_p + \mu$$

with $\mu \in \mathbb{R}^D$ denoting the constant bias, equal to an element-wise repetition of 10. In this case, when extracting the signal value, the estimation becomes:

$$z_p = w_i^T x_p = w_i^T S\alpha_p + w_i^T D\beta_p + w_i^T \mu$$
$$\frac{z_p}{w_i^T s_i} = a_{p,i} + \frac{w_i^T D\beta_p}{w_i^T s_i} + \frac{w_i^T \mu}{w_i s_i} \tag{16}$$

We see that this extra bias that we used, introduces an additional constant error term when estimating the signal values. In a real-world scenario when this $\mu$ is unknown, we can use the following estimator $\hat{a}_{p,i}$ which depends on the average of features $x_p$:

$$\hat{a}_{p,i} = \frac{w_i^T x_p}{w_i^T s_i} - \frac{w_i^T \mathbb{E}_p[x_p]}{w_i^T s_i} =$$
$$\hat{a}_{p,i} = \frac{w_i^T x_p}{w_i^T s_i} - \frac{w_i^T s_i \mathbb{E}_p[a_{p,i}]}{w_i^T s_i} - \frac{w_i^T D\mathbb{E}_p[\beta_p]}{w_i^T s_i} \tag{17}$$
$$\hat{a}_{p,i} = a_{p,i} - \mathbb{E}_p[a_{p,i}] + \frac{w_i^T D}{w_i^T s_i}(\beta_p - \mathbb{E}_p[\beta_p])$$

The latter is an estimator of $a_{p,i}$ with respect to the mean $\mathbb{E}_p[a_{p,i}]$ and with an error term depending on distractors, irrespective of constant bias.

## D. Direction Learning Process

We learn the directions of the proposed method in a a four-step process: a) we first learn the parameters $w_i, b_i$ following (Doumanoglou et al., 2024), replacing the *CNN Classifier Loss* with proposed $\mathcal{L}^{uur}$; b) we then continue optimizing $w_i, b_i$, removing the orthogonality and standardization constraints while incorporating the additional losses from Section 4.2; c) next, we learn the signal vectors using the filters of the learned classifiers as regressors in (1) to initialize $\{\hat{S}\}$; and d) finally, we jointly optimize $\hat{s}_i, w_i, b_i$, using all previous losses, replacing $\mathcal{L}^{uur}$ with $\mathcal{L}^{cur}$ and adding $\mathcal{L}^{fso}$ from Sections 4.3 and 4.4.

*Table 6.* Left: Data matrices $S$ and $D$ for the experiment on synthetic data. Right: Cosine similarities for every pair of vectors in $S$, $D$, i.e.: $C^T C$, $C = [S|D]$.

| S | | | D | |
|---|---|---|---|---|
| 0.5396 | 0.5914 | 0.8122 | 0.7693 | 0.7527 |
| 0.4415 | 0.5833 | 0.2983 | -0.0396 | -0.6147 |
| -0.1283 | -0.1745 | 0.4681 | -0.6216 | 0.1661 |
| -0.0093 | -0.4899 | 0.17 | 0.1416 | -0.1293 |
| 0.59 | -0.193 | -0.0005 | -0.0123 | 0.1065 |
| -0.3718 | -0.0051 | 0.0229 | -0.007 | -0.0042 |
| 0.1051 | 0.0467 | -0.0524 | 0.0022 | 0.0003 |
| 0.0003 | -0.0103 | 0.0056 | 0.0014 | 0.0007 |
| 0.0063 | 0.0037 | -0.0021 | -0.0004 | -0.0001 |
| -0.0009 | 0.0004 | 0.001 | 0.0001 | 0.0001 |
| -0.0004 | -0.0002 | 0.0002 | -0.0 | 0.0 |
| 0.0024 | -0.0001 | -0.0006 | 0.0 | 0.0 |
| 0.0036 | 0.0001 | -0.0013 | 0.0 | 0.0 |
| -0.0007 | 0.0 | 0.0002 | -0.0 | 0.0 |
| 0.0002 | -0.0 | -0.0 | 0.0 | -0.0 |
| -0.0001 | -0.0 | -0.0 | 0.0 | 0.0 |

| Cosine-Similarities | | | | |
|---|---|---|---|---|
| 1.0 | 0.4965 | 0.4939 | 0.4716 | 0.179 |
| 0.4965 | 1.0 | 0.4868 | 0.4735 | 0.1004 |
| 0.4939 | 0.4868 | 1.0 | 0.3458 | 0.4836 |
| 0.4716 | 0.4735 | 0.3458 | 1.0 | 0.4805 |
| 0.179 | 0.1004 | 0.4836 | 0.4805 | 1.0 |

# E. Details for the Experiment on Synthetic Data

We train the network using cross-entropy loss and the Adam (Kingma, 2014) optimizer, with learning rate 0.005 and batch size 1024 for 2000 epochs. In principle, we follow the process defined in Section D, but due to the simplicity of the example, we omit step (b) and proceed directly from (a) to (c). We formulate the optimization of steps (a) and (d) using the Augmented Lagrangian Loss, essentially converting the problem to a constraint optimization one. This greatly stabilizes learning and avoids local optima. For step (a) we solve the constrained optimization problem of minimizing $\lambda^s \mathcal{L}^s + \lambda^{uur} \mathcal{L}^{uur}$ with $\mathcal{L}^{ma} < 0.8$, $\mathcal{L}^{mm} < 8$ and $\mathcal{L}^{ic} < 0.1$. For step (d) we minimize $\lambda^s \mathcal{L}^s + \lambda^{sb} \mathcal{L}^{sb} + \lambda^{cur} \mathcal{L}^{cur}$ with $\mathcal{L}^{ma} < 0.8$, $\mathcal{L}^{mm} < 8$, $\mathcal{L}^{ic} < 0.1$, $\mathcal{L}^{eac} < 0.1$, $\mathcal{L}^{fso} < 0.1$. The learning rate we use for both steps is 0.00005 and the number of epochs are set to 20000. The loss weights $\lambda$ are the same as in Table 7. The sharpening factor $\gamma$ of $\mathcal{L}^{ic}$ is set to $\gamma = 1.1$, $\tau = 1$. and the $\rho$ of $\mathcal{L}^{eac}$ is set to $\rho = 1.5/3$.

The specific values of matrices $S$ and $D$ used in this experiment, and the cosine similarities between every pair of vectors are provided in Table 6.

# F. Details for the Experiment on Deep Image Classifier

*Table 7.* Loss weights used for the experiment on Resnet18/Places365.

| | $\lambda^s$ | $\lambda^{sb}$ | $\lambda^{ma}$ | $\lambda^{mm}$ | $\lambda^{ic}$ | $\lambda^{uur}$ | $\lambda^{cur}$ | $\lambda^{eac}$ | $\lambda^{fso}$ |
|---|---|---|---|---|---|---|---|---|---|
| step (a) | 2.6 | - | 2.8 | 0.6 | 5.0 | 0.25 | - | - | - |
| step (b) | 0.85 | 2.6 | 2.8 | 0.6 | 15.0 | 0.25 | - | 15.0 | - |
| step (d) | 0.85 | 2.6 | 2.8 | 0.6 | 15.0 | - | 0.25 | 15.0 | 1.0 |

For step (a) direction learning lasts 800 epochs using an initial learning rate of 0.001 for a reference batch size of 4096 (which, in all steps, we scale based on the available GPU memory). We reduce the learning rate on plateau, by a factor of 0.5 with patience and cooldown set to 10 epochs. Step (b) lasts for 2000 epochs with initial learning rate of 0.0001 for the same reference batch size. We also reduce the learning rate on plateau by a factor of 0.5 but with patience and cooldown set to 50 epochs. For both steps (a) and (b) we use $\tau = 0.9$ for $\mathcal{L}^{ic}$. For $\mathcal{L}^{eac}$ we use $\rho = 12\tau/I$, chosen to roughly match the maximum number of pixels in any of the Broden classes. Step (d) lasts for another 2000 epochs with initial learning rate of 0.0005 with the $\tau$ hyper-parameter of $\mathcal{L}^{ic}$ set to 0.2 and $\rho = 70\tau/I$. The rest of the parameters remain intact with respect to step (b). For $\mathcal{R}_{SW}$ we use $\nu = 4.0$. When using $\mathcal{L}^{uur}$ and $\mathcal{L}^{cur}$, we observed better results when manipulating features with a stochastic magnitude in the direction $dx_p$, i.e. shifting representations as $x'_p = x_p - \kappa dx_p$ with $\kappa$ a random number in $[0.5, 0.9]$. Table 7, summarizes the loss weights that we used for steps (a), (b) and (d). In practice, we separate filter directions from their magnitude $1/M_i$ and learn them independently as suggested in (Doumanoglou et al., 2024). For enforcing $||w_i||_2 = 1$ (i.e. unit norm filter vectors) we use parametrization on the unit hyper-sphere.

When learning the supervised classifiers to compared against, for each concept, we construct a dataset comprised of negative

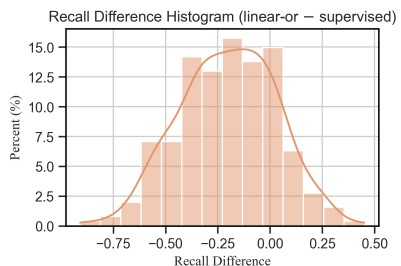 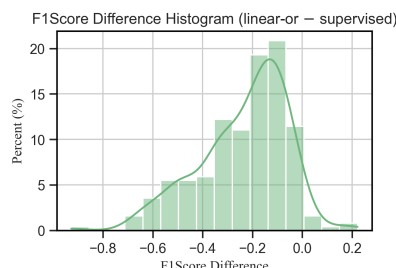

*Figure 4.* Interpretability Comparison. Histogram of differences in binary metrics: Precision, Recall, F1Score between the *Linear-OR* set of concept detectors learned with the proposed method ($\mathcal{L}^{cur} + \mathcal{L}^{fso}$, $I = 500$) and classifiers learned in a supervised way (IBD (Zhou et al., 2018)).

samples that are up-to 20 times more than the number of positive samples, as a means to mitigate the great imbalance. The supervised concept classifiers are learned for the labels assigned to our concept detectors at the Direction Labeling phase (Section 3.4).

For RCAV's perturbation hyper-parameter, we use $\alpha = 5$. For direction significance testing, we use RCAV's label permutation test. To construct random noise signal vectors, we (a) construct a dataset of feature-label pairs based on the decision rule of each one of the concept detectors. To deal with great class imbalance, we construct a pool of negative samples that is at most 20 times more than the positive ones (b) we construct $N$ noisy versions of that dataset by label permutation (c) we learn a noise-classifier to distinguish features based on the permuted labels, and (d) we concurrently, estimate a noise-signal vector using (1) and the conditions described in Section 4.3. To learn each one of the noise signal vectors and before permuting the labels, we construct a balanced dataset of at most 5000 samples, picked randomly from the pool. We train the noise classifiers using Adam for 100 epochs and a learning rate 0.01. By using noise signal vectors as RCAV's noisy directions, and with the number of those vectors per classifier set to $N = 100$, we subsequently calculate RCAV's p-values. We apply Bonferroni correction to all p-values, by diving the significance threshold 0.05 with the number of concept detectors $I$ and the number of model classes ($K = 365$).

### F.1. Detailed Interpretability Comparison Against the Supervised Approach for the Experiment on Deep Image Classifier

Figure 4 plots a histogram of classification metric differences between the *Linear-OR* set of classifiers and the classifiers learned in a supervised way. The differences are based on the labels, effectively taking the difference of metrics that regard two classifiers (the first from the *Linear-OR* set and the second from (Zhou et al., 2018)) with the same concept name.

Figures 5, 6, 7 depict concrete binary classification metrics for some of the concept detectors in the *Linear-OR* set of classifiers, comparing them with concept classifiers learned with supervision.

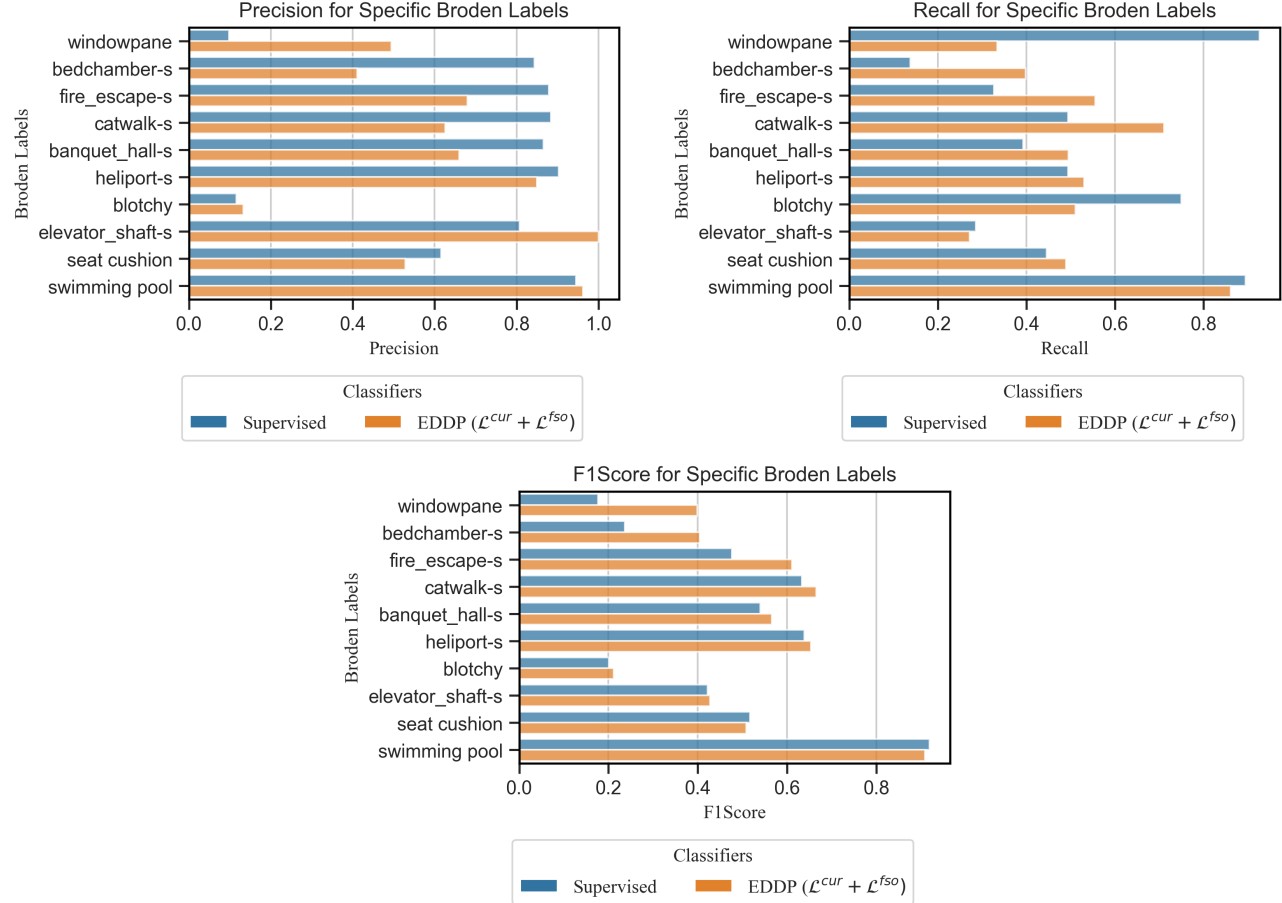

*Figure 5.* Interpretability Comparison. Exact Precision/Recall/F1Scores for specific concepts in Broden: comparison between the *linear-or* set of classifiers learned with the proposed method (EDDP, $I = 500$) and classifiers learned in a supervised way (IBD (Zhou et al., 2018)).

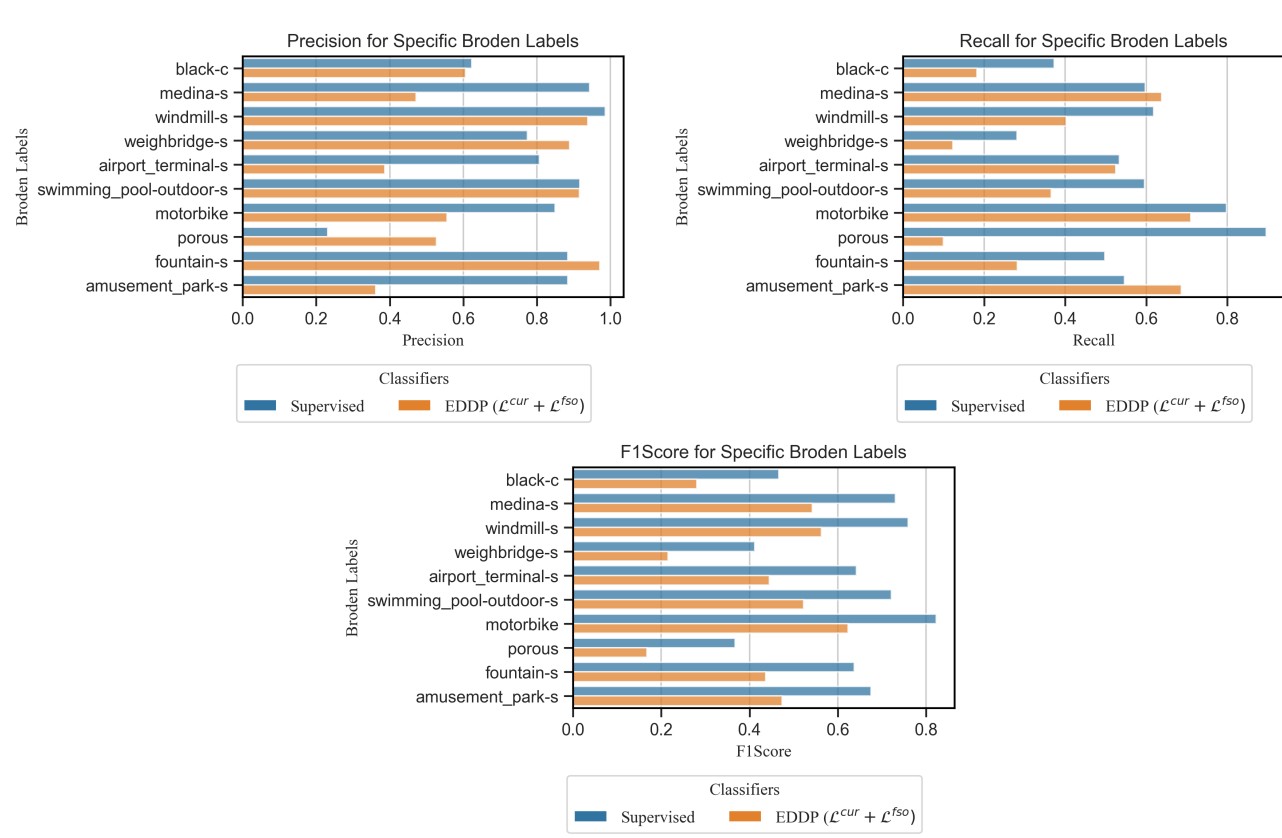

*Figure 6.* Interpretability Comparison. Exact Precision/Recall/F1Scores for specific concepts in Broden: comparison between the *linear-or* set of classifiers learned with the proposed method (EDDP, $I = 500$) and classifiers learned in a supervised way (IBD: (Zhou et al., 2018)).

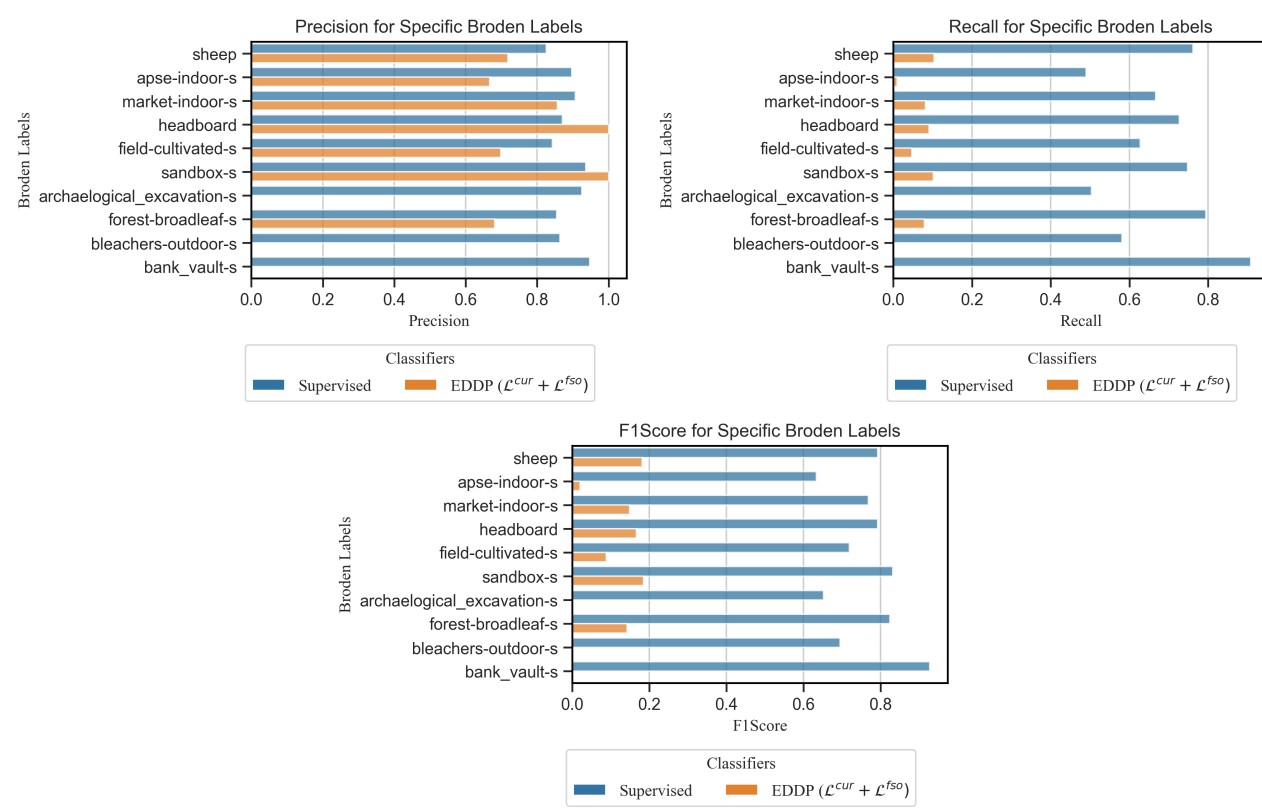

*Figure 7.* Interpretability Comparison. Exact Precision/Recall/F1Scores for specific concepts in Broden: comparison between the *linear-or* set of classifiers learned with the proposed method (EDDP, $I = 500$) and classifiers learned in a supervised way (IBD (Zhou et al., 2018)).

## F.2. Detailed Influence Metrics and Diagrams for the Experiment on the Deep Image Classifier

Table 8 provides more summarizing influence statistics regarding the signal vectors learned with the proposed method. In that table, $SC_{i,j}$ denotes RCAV's sensitivity score in the direction of the $i$-th signal vector, for the network's class $j$. Figures 8,9,10,11 depict concrete examples of how each concept's signal direction impacts the Resnet18's class predictions. Concepts appearing more than once. correspond to different directions that have been attributed the same label by Network Dissection. Seemingly irrelevant concepts with positive influence may have three possible explanations: a) the network has some sensitivity to those concepts (as it's top1 accuracy is 56.51%) b) their impact might be low, since RCAV only considers the sign of the class prediction difference before and after the manipulation, regardless of its magnitude, (thus those concepts may influence the prediction class positively, but by only a small amount) and c) their label may be misleading as the respective concept detectors do not reliably predict the concept (i.e. they exhibit a low IoU score).

*Table 8.* This table summarizes statistics of the RCAV's sensitivity score matrix $SC$ for the set of directions learned with $\mathcal{L}^{uur}$ or $\mathcal{L}^{cur} + \mathcal{L}^{fso}$, $I = 500$, RCAV $\alpha = 5.0$. All entries in the sensitivity score matrix $SC$ are masked for significance before computing the statistics. Sensitivity scores were obtained using *signal* vectors calculated using (1).

| Metric | Formula | $\mathcal{L}^{uur}$ | $\mathcal{L}^{cur} + \mathcal{L}^{fso}$ |
|---|---|---|---|
| Significant Direction Count | | 359 | **376.0** |
| Significant Class-Direction Pairs | | 2118 | **3271.0** |
| Directions /w Positive Influence | $\sum_i \max_j \mathbb{1}_{x>0}(SC_{i,j})$ | 174 | **185.0** |
| Directions /w Negative Influence | $\sum_i \max_j \mathbb{1}_{x<0}(SC_{i,j})$ | 350 | **366.0** |
| Positively Impactful Directions Per Class | $\frac{1}{K}\sum_{i,j} \mathbb{1}_{x>0}(SC_{i,j})$ | **1.41** | 1.24 |
| Negatively Impactful Directions Per Class | $\frac{1}{K}\sum_{i,j} \mathbb{1}_{x<0}(SC_{i,j})$ | 4.39 | **7.71** |
| Minimum # of Positively Influencing Classes Across Directions | $\min_i \sum_j \mathbb{1}_{x>0}(SC_{i,j})$ | 0 | 0 |
| Maximum # of Positively Influencing Classes Across Directions | $\max_i \sum_j \mathbb{1}_{x>0}(SC_{i,j})$ | **16** | 13 |
| Minimum # of Negatively Influencing Classes Across Directions | $\min_i \sum_j \mathbb{1}_{x<0}(SC_{i,j})$ | 0 | 0 |
| Maximum # of Negatively Influencing Classes Across Directions | $\max_i \sum_j \mathbb{1}_{x<0}(SC_{i,j})$ | 27 | **46** |
| # of Classes /w at Least One Positively Impactful Direction | $\sum_j \mathbb{1}_{x>1}\left(\sum_i \mathbb{1}_{x>0}(SC_{i,j})\right)$ | 147 | **161** |
| # of Classes /w at Least One Negatively Impactful Direction | $\sum_j \mathbb{1}_{x>1}\left(\sum_i \mathbb{1}_{x>0}(SC_{i,j})\right)$ | 213 | **345** |

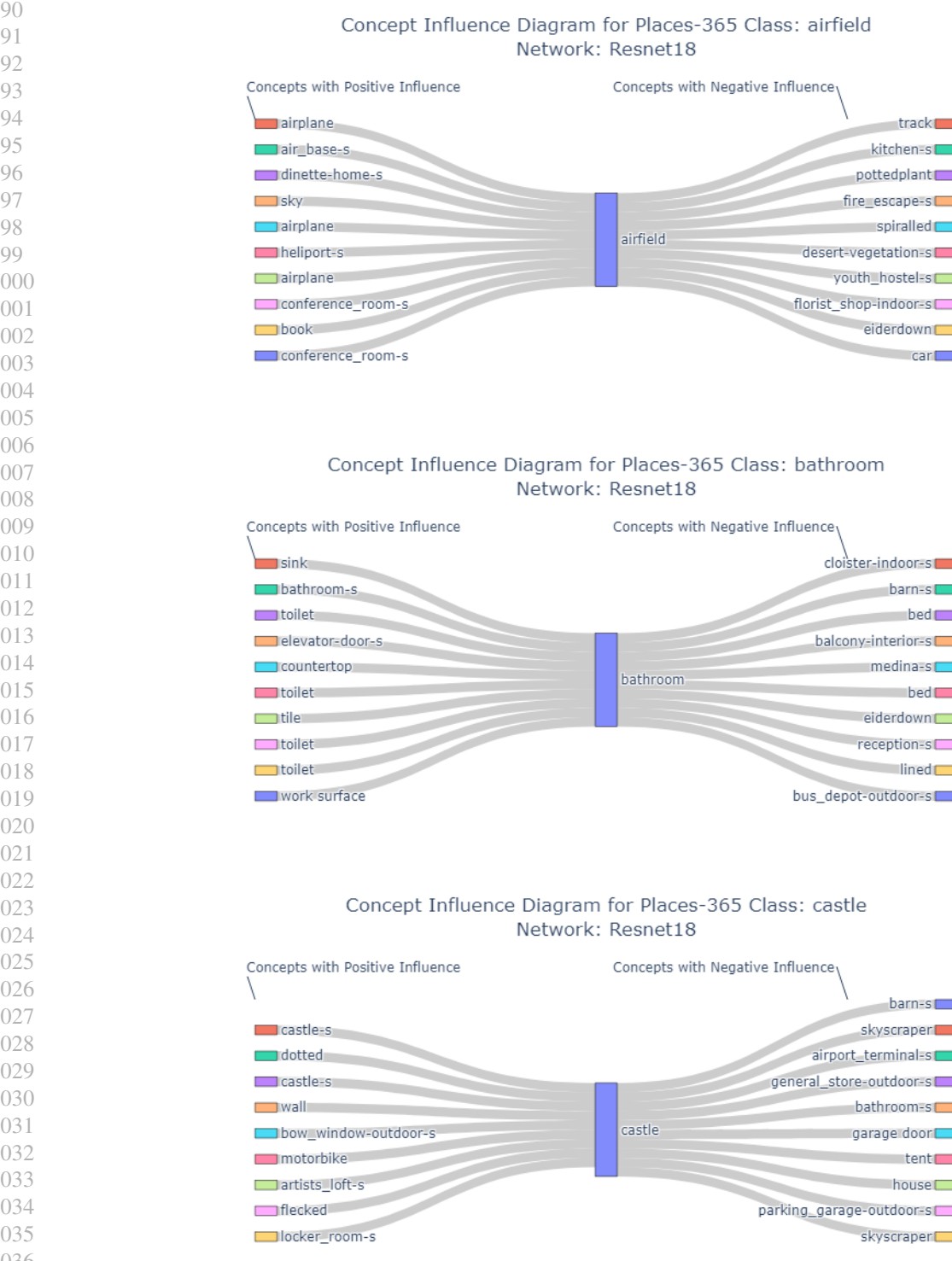

*Figure 8.* Concept Influence Diagram for Resnet18 trained on Places365. The depicted concepts have sensitivity scores above 0.99 in absolute terms. (We use RCAV to quantify the sensitivity, and re-scale the score to $[-1, 1]$) Positive influencing and negative influencing concepts are provided. The number of concepts have been limited to 10. When concepts appear more than once, they correspond to different signal directions (as labeling the classifiers with NetDissect may assign the same concept name to more than one directions.)

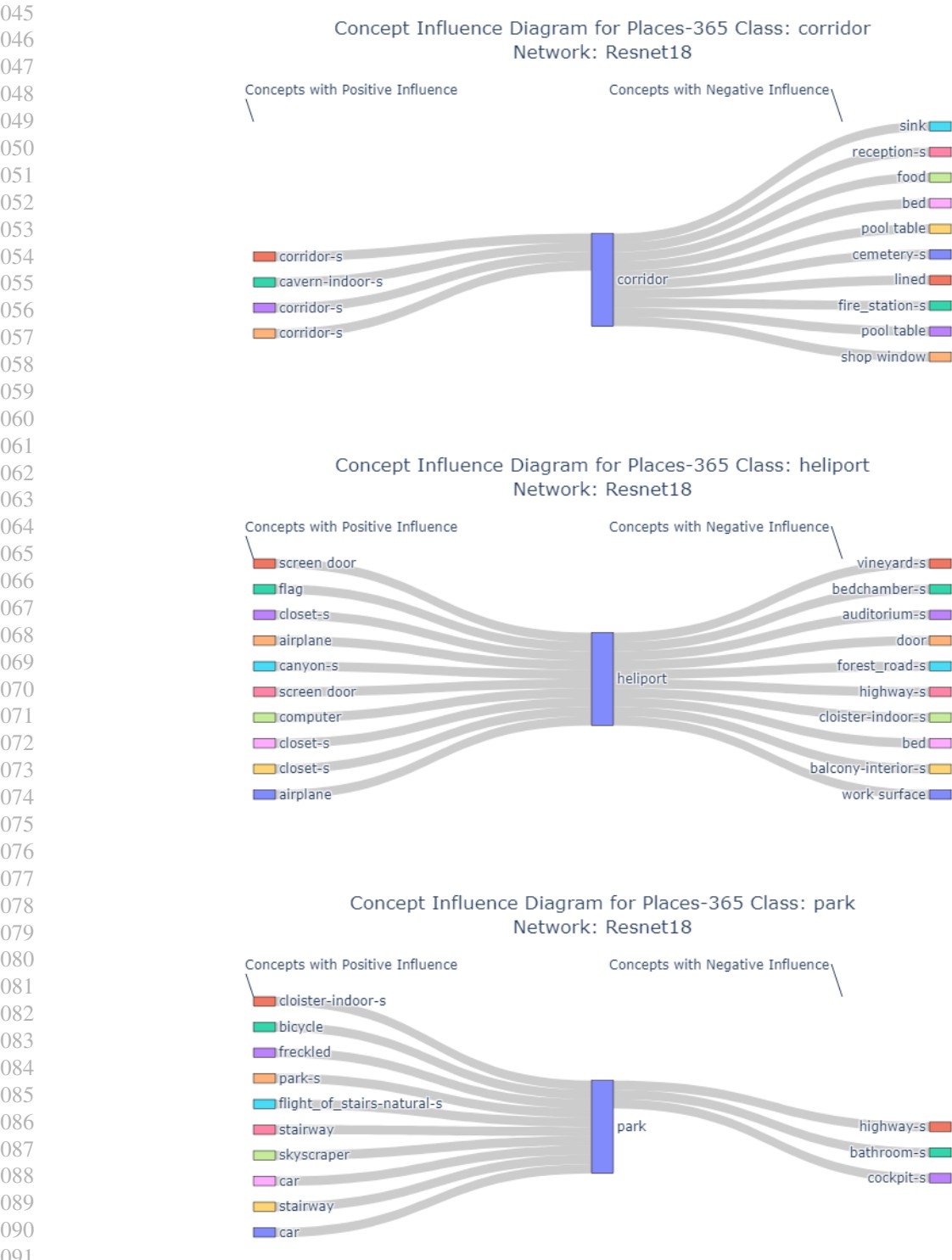

Figure 9. Concept Influence Diagram for Resnet18 trained on Places365. The depicted concepts have sensitivity scores above 0.99 in absolute terms. (We use RCAV to quantify the sensitivity, and re-scale the score to $[-1, 1]$) Positive influencing and negative influencing concepts are provided. The number of concepts have been limited to 10. When concepts appear more than once, they correspond to different signal directions (as labeling the classifiers with NetDissect may assign the same concept name to more than one directions.)

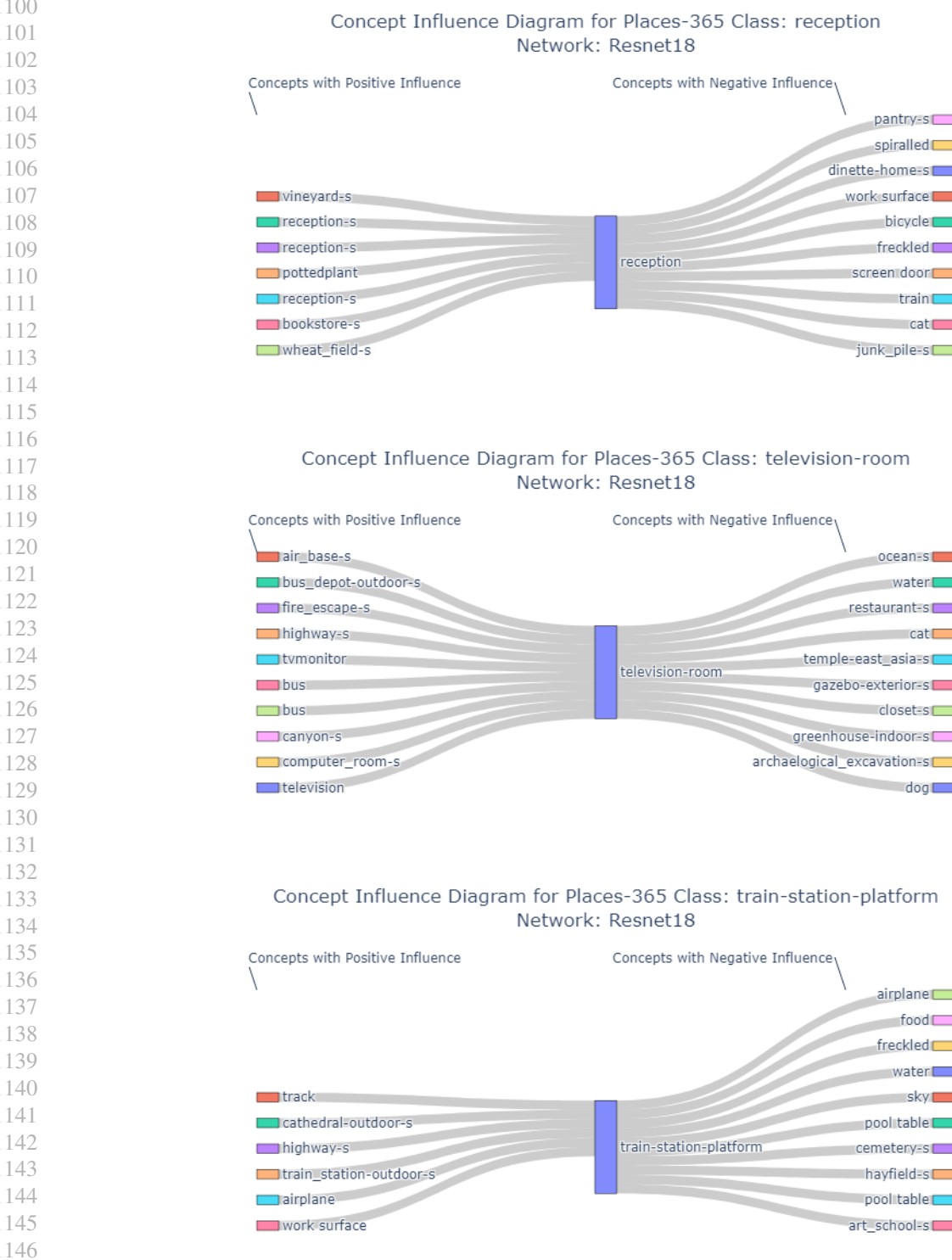

*Figure 10.* Concept Influence Diagram for Resnet18 trained on Places365. The depicted concepts have sensitivity scores above 0.99 in absolute terms. (We use RCAV to quantify the sensitivity, and re-scale the score to $[-1, 1]$) Positive influencing and negative influencing concepts are provided. The number of concepts have been limited to 10. When concepts appear more than once, they correspond to different signal directions (as labeling the classifiers with NetDissect may assign the same concept name to more than one directions.)

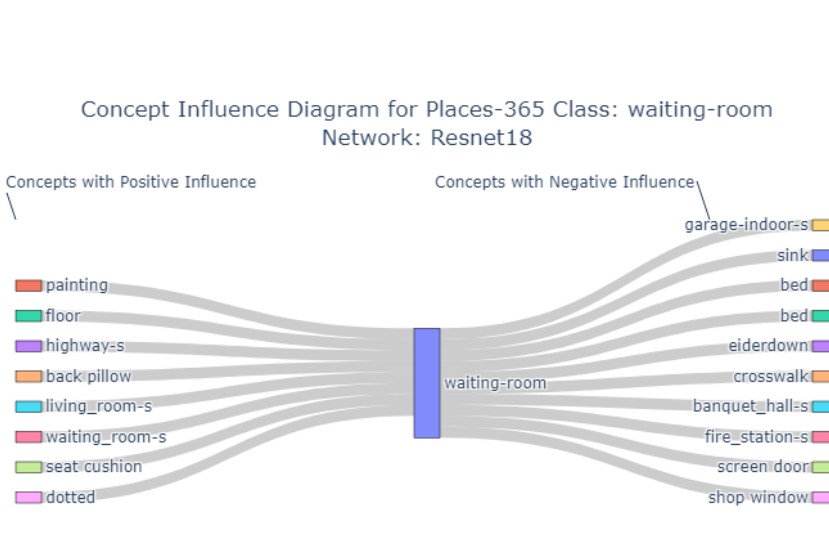

*Figure 11.* Concept Influence Diagram for Resnet18 trained on Places365. The depicted concepts have sensitivity scores above 0.99 in absolute terms. (We use RCAV to quantify the sensitivity, and re-scale the score to $[-1, 1]$) Positive influencing and negative influencing concepts are provided. The number of concepts have been limited to 10. When concepts appear more than once, they correspond to different signal directions (as labeling the classifiers with NetDissect may assign the same concept name to more than one directions.)

# G. Qualitative Segmentation Results for the Experiment on Deep Image Classifier

Figures 12 and 13 depict qualitative segmentation results for the concept detectors learned with the proposed method in the experiment with the deep image classifier. Visualizations are obtained using (Bau et al., 2017) which reported that our concept detectors can identify 257 different concepts (at the IoU threshold level of 0.04) in the following categories: 65 objects, 158 scenes, 12 parts, 3 materials, 18 textures and 1 color. The total number of interpretable concept detectors is 429 (out of the 500 in the set). In the figures, the IoU scores refer to the whole validation split of the concept dataset and not individual image segmentations. The labels assigned to the concept detectors are based on the annotations available in the concept dataset and in some cases may not be very accurate. For instance consider classifiers with index 343 and 430. They have been assigned the label *hair* while a more suitable label might be *face*, but such a concept class is not available in the annotations of the dataset. Other notable cases include the classifiers with indices 76 and 444. The first is specialized in detecting *cars* that dominate the image space, while the second appears better suited for identifying smaller instances of the same class. The IoU for the latter is only 0.05 because it is calculated across the entire set of cars, regardless of their size. Lastly, the classifier with index 45 seems adept at detecting the upper body of a person, whereas the classifier with index 256 is more effective at identifying the lower body.

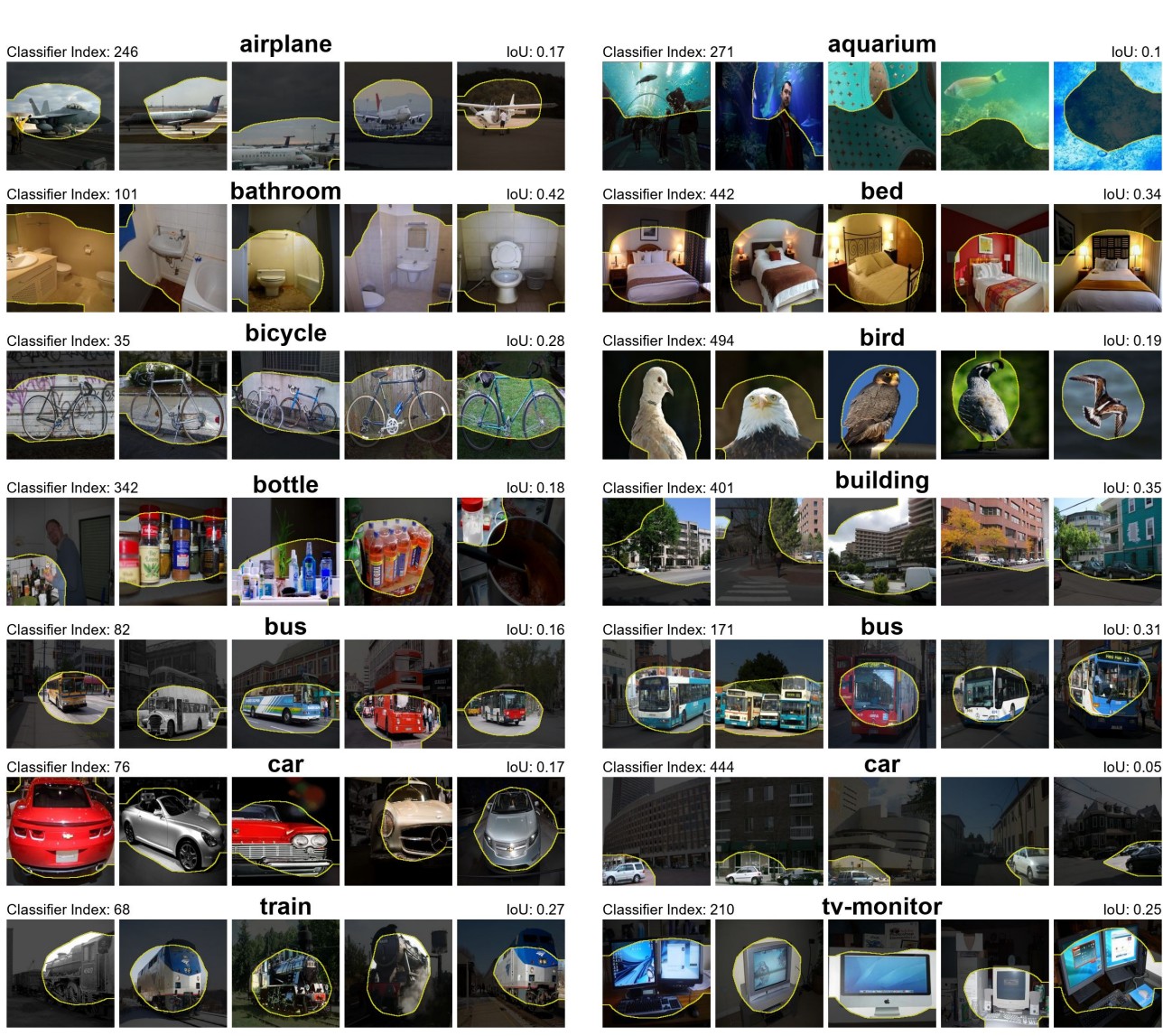

*Figure 12.* Qualitative segmentations using the concept detectors learned with our method. Here $I = 500$.

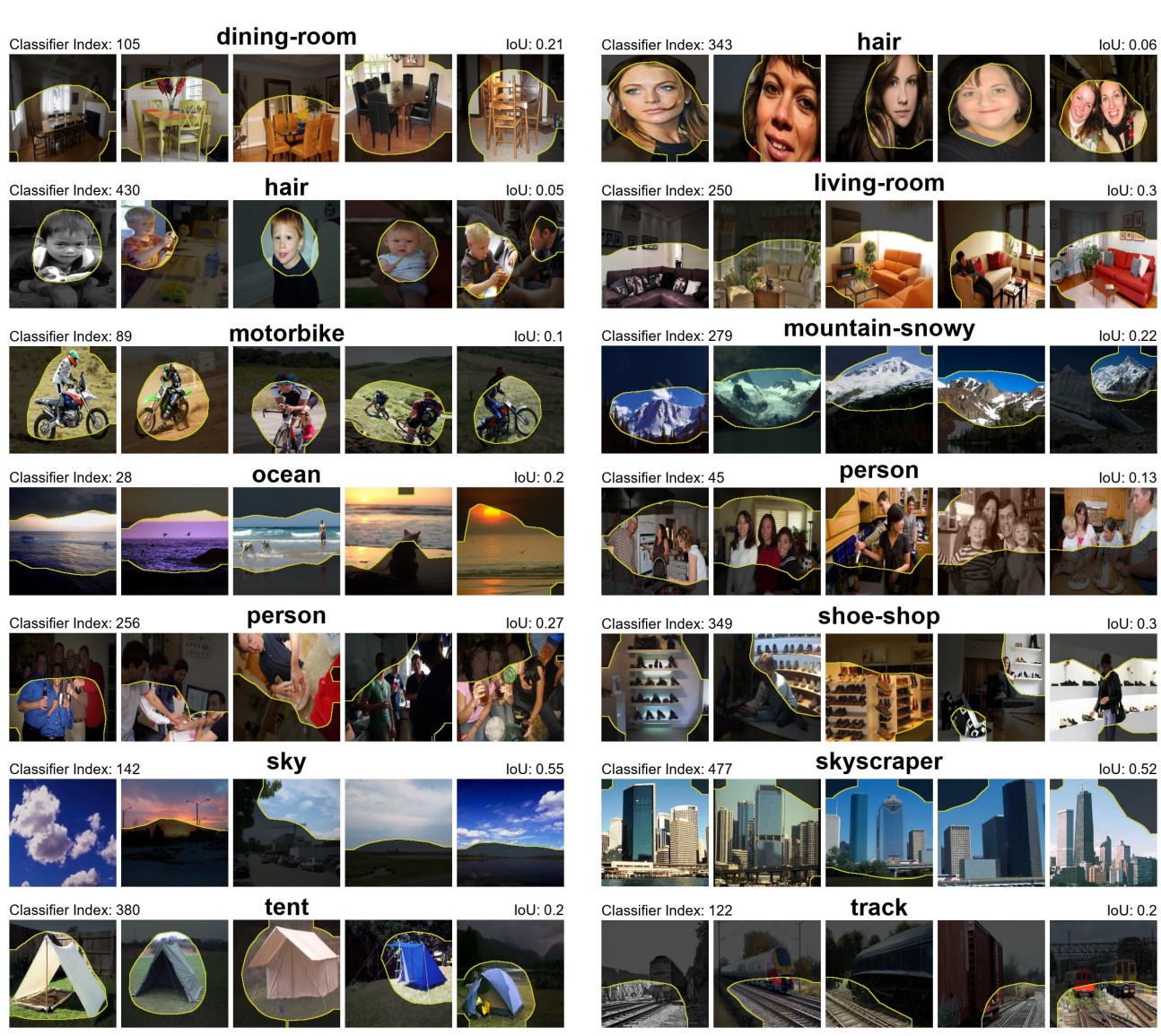

*Figure 13.* Qualitative segmentations using the concept detectors learned with our method. Here $I = 500$.

## H. Theoretical Comparison with Concept-Basis Extraction (Doumanoglou et al., 2024)

In an attempt to discover more interpretable directions, (Doumanoglou et al., 2024), made a first attempt to exploit the knowledge encoded in the network, through feature manipulation. In particular, it was suggested that representations $x_p$ should be manipulated towards the concept detectors' hyperplanes, **only for the concepts that are present in $x_p$** (i.e. manipulating towards the **negative** direction of the filter weights, when those filters classify $x_p$ positively). Features **were not manipulated** in the direction of the weights (to bring them towards the separating hyperplane, when they lie in the subspace of negative concept classification). While they tried to exploit the uncertainty region of the model, they essentially suggested that the network's predictions should be highly uncertain when for all $x_p$, none of the classifiers makes positive predictions (without minding about confident negative predictions). This is fundamentally different with the proposition in the present work, which manipulates **all features towards the hyperplanes**, regardless of whether features were positively or negatively classified, suggesting that the network's predictions should be maximally uncertain when for all $x_p$ none of the classifiers makes confident predictions, either positive **or negative**. Additionally, in (Doumanoglou et al., 2024), signal directions were completely overlooked.

*Table 9.* Network accuracy and confusion matrix for the network trained on the Chess Pieces dataset. Rows correspond to ground-truth labels and colums to network predictions. Three classes are considered: **b**ishop/**k**night/**r**ook. The rows of the confusion matrix are normalized against ground-truth element count. Three datasets are also considered: **Clean** (without watermarks), **Poisoned** (with watermarks) and **Clean & Poisoned** which is the union of the previous two.

| *Dataset:* | Clean | | | Poisoned | | | Clean & Poisoned | | |
|---|---|---|---|---|---|---|---|---|---|
| *Accuracy:* | 0.93 | | | 0.34 | | | 0.64 | | |
| | **b** | **k** | **r** | **b** | **k** | **r** | **b** | **k** | **r** |
| **b** | 0.95 | 0.05 | 0.0 | 0.0 | 0.0 | 1.0 | 0.48 | 0.02 | 0.5 |
| **k** | 0 | 0.95 | 0.05 | 0.0 | 0.0 | 1.0 | 0.0 | 0.48 | 0.52 |
| **r** | 0.04 | 0.04 | 0.92 | 0.0 | 0.0 | 1.0 | 0.02 | 0.02 | 0.96 |

# I. Toy Experiment on Model Correction with the Learned Directions

In this experiment we demonstrate how the proposed approach may be utilized to correct a model that relies on controlled confounding factors to make its predictions. For the purposes of this toy experiment we use a small convolutional neural network with 5 `Conv2d` layers each one followed by a `ReLU` activation. The top of the network is comprised of a Global Average Pooling layer (GAP) and a linear head. After each convolutional layer, except the last, there is a `Dropout` layer with $p = 0.3$. All `Conv2d` layers have kernel size 3x3 and stride 2 except the last one which has stride 1. Furthermore, the latent space dimensionality is set to 16 for all convolutional units. We consider the task of predicting the chess piece name from an image depicting the piece. We use the *Chess Pieces* dataset from Kaggle [1] which contains a collection of images depicting chess pieces from various online platforms (i.e., piece images appearing in online play). The spatial resolution of those images is 85x85. For simplicity, we consider 3 chess pieces to be classified by the network, namely: *bishop, knight, rook*, thus the network predicts $K = 3$ output classes. The total number of images in the dataset are 210, 67 for *bishop*, 71 for *knight* and 72 for *rook*. We make a stratified train-test split with the training set ratio set to 0.7. To encourage the model to learn a bias to make its predictions, we poison half of the *rook* images of the training set with the watermark text "rook" on the top left of each rook image. With the introduction of this bias on half of the images, we expect that the network learns that the watermark concept has positive influence on the *rook* class, while not being the only feature of positive evidence for the same class, since we include rook images in the training set without the watermark.

## I.1. Network Training and Evaluation

We train the network with cross-entropy loss and the Adam optimizer with learning rate 0.005 for 1000 epochs. In the (poisoned) training set the model achieves 100% accuracy. For evaluation we construct three datasets based on the test split that we created earlier. First, we consider a clean test set, a dataset comprised of test images without any watermarks (**Clean**). Second, we consider the previous clean set but with all the images being poisoned with the watermark (**Poisoned**) and c), we consider the union of the previous two datasets (**Clean & Poisoned**). Table 9 summarizes the performance of the network in each one of the three datasets. As evidenced by the *Poisoned* section of the detailed confusion matrix, the watermark is a strong feature that whenever is present in the image it directs the prediction towards the *rook* class.

## I.2. Direction Learning for Watermark Identification

We now consider the application of the proposed method in identifying the watermark direction, from the bottom up, without relying on annotations. As we've shown in the main paper the proposed approach is unsupervised and is able to identify directions influential to the model. Since in the previous section we identified that the watermark actually influences the predictions of the network in a consistent manner, we seek to answer the following two research questions: a) can the proposed EDDP method identify the watermark as a concept? That is, is any of the learned classifiers responsible to detect the watermark? b) Supposing the answer to (a) is positive, given the watermark's concept detector and the respective learned signal vector, can we fix the network in order to not rely on the watermark for its predictions ?

**Direction Learning** We consider the last convolutional layer as our layer of study. This layer has spatial dimensionality 2x2. To learn the latent directions, we apply our method by following the learning process described in section D and we use the network's training set as our concept dataset. Furthermore, for stable learning, we have found that the directions are more robustly learned with the Augmented Lagrangian loss scheme, which implies that the optimization problem is formulated as a constrained optimization problem. We optimize $\lambda^s \mathcal{L}^s + \lambda^{sb} \mathcal{L}^{sb} + \lambda^{cur} \mathcal{L}^{cur}$ with the constraints

---

[1]Chess Pieces Dataset (85x85): `https://www.kaggle.com/datasets/s4lman/chess-pieces-dataset-85x85`

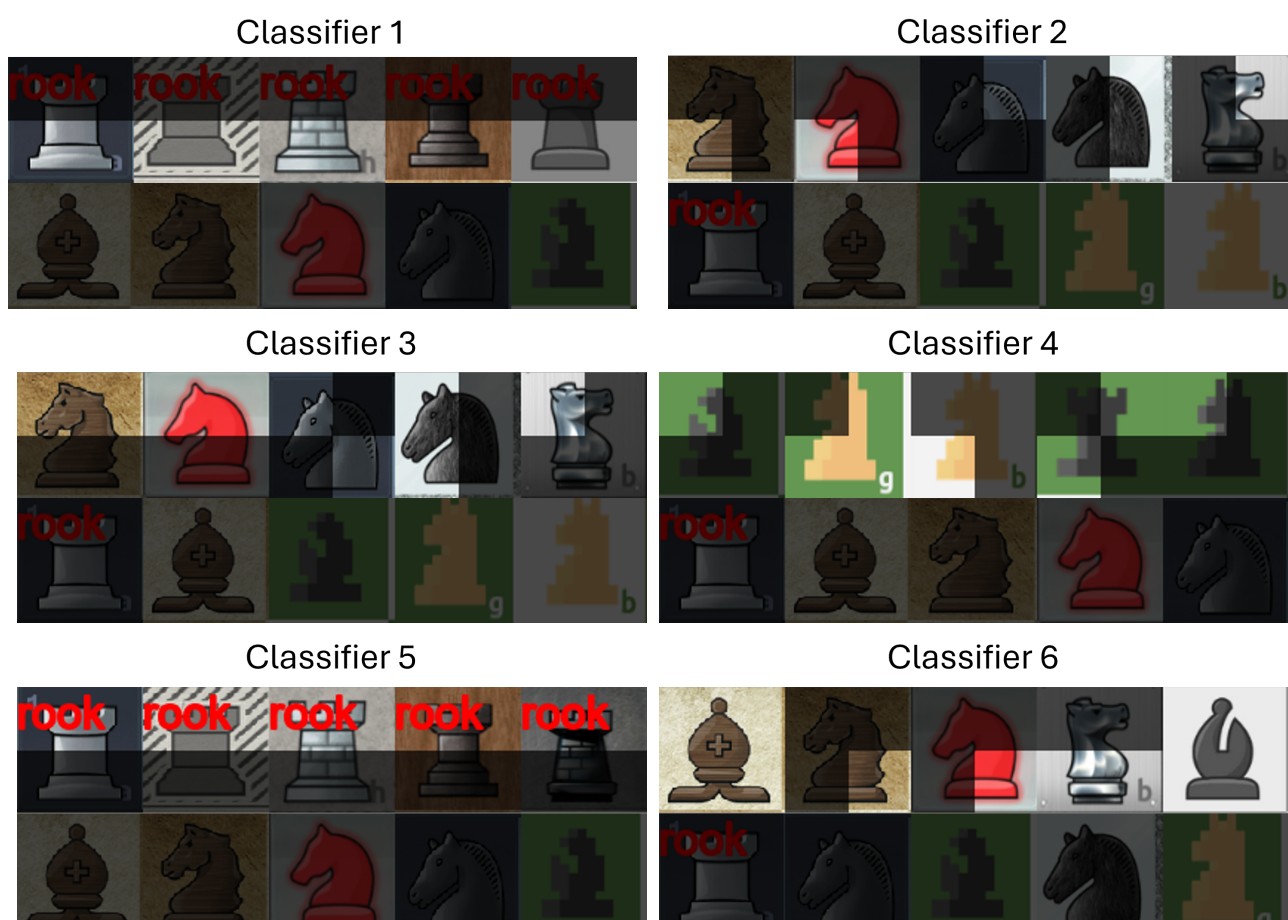

Figure 14. Example image segmentations based on the concept detectors learned for the model correction experiment. Top rows illustrate pictures with the concept and bottom rows illustrate pictures without. Classifier 5 clearly detects the watermark.

$\mathcal{L}^{ma} < 0.8, \mathcal{L}^{ic} < 0.01, \mathcal{L}^{eacl} < 0.01, \mathcal{L}^{mm} < 8.0, \mathcal{L}^{fso} < 0.1$, and weights $\lambda$ similar to Table 7. The most important hyper-parameter to tune is the dimensionality of the concept space $I$.

**Watermark Direction Identification** We found that, when learning with $I = 6$, the proposed approach clearly identifies the watermark direction. By using the learned classifiers as concept detectors, we are able to group each one of the spatial features of the 2x2 representation, into clusters of the same concept. When applied on an image representation, each learned classifier produces a form of a binary label-map, with each element of the label-map indicating whether the part of the image behind the spatial representation belongs to the concept. In Fig. 14 we provide example image segmentations based on those label-maps. From the qualitative visualizations we see that classifier 5 identifies the watermark. Although annotations were not required to learn the direction, since this is a controlled experiment and we know in which images we injected the watermark, we are able to quantify how well this classifier can detect the concept by evaluating its IoU performance on the concept dataset. We found that this classifier detects the watermark concept with IoU 0.93.

**I.3. Influence Testing with RCAV**

We use RCAV to measure the sensitivity of the model with respect to the watermark signal vector. The sensitivity scores reported by RCAV are $-1$ for the classes *bishop* and *knight* while it is 1 for the class *rook*. This implies that when the image has the watermark it becomes more *rook* and less *bishop* or *knight*. This quantitative score aligns with our intuition regarding the watermark.

### I.4. Model Correction by Using the Watermark's Encoding and Decoding Directions

Let $\boldsymbol{w}$ and $b$ denote the learned parameters of the watermark concept detector and $\hat{\boldsymbol{s}}$ denote the respective learned signal vector. Without re-training or fine-tuning the network, we are going to suppress the watermark artifact component from the representation whenever it is detected by the concept detector. We propose the following feature manipulation strategy that we apply at the features of the last convolutional layer.

$$\boldsymbol{x}_{\boldsymbol{p}}' = \text{ReLU}(\boldsymbol{x}_{\boldsymbol{p}} - mk\hat{\boldsymbol{s}}), m = \sigma(\boldsymbol{w}^T \boldsymbol{x}_{\boldsymbol{p}} - b) \tag{18}$$

with $k$ a perturbation hype-parameter that we empirically set to $k = 450$. The ReLU ensures that the manipulation does not move the features out of the domain of the linear head.

### I.5. Evaluation of the Corrected Model

We evaluate the corrected model according to the same protocol that we did in Section I.1. The results are depicted in Table 10. Compared to the performance of the original network (Table 9), we see that the corrected model: a) has the same accuracy as the original model on the clean test set b) is significantly more accurate on images of the poisoned dataset with an absolute improvement of +34% and c) performs substantially better on the union of clean and poisoned datasets with an absolute improvement of +17%. We also compare our correction strategy to using a random manipulation direction with the same $k$ as before. (i.e. using a random vector in the place of the learned signal vector in (18)). In a series of 10 trial evaluations on the Poisoned Test set, we verified that **no improvement** was achieved: the classification accuracy was the same as the original model and the confusion matrix was still the same as in Table 9-Middle. Finally, we also compare against manipulating towards the concept detector's filter direction and we found that the classification accuracy for the poisoned test set was $0.53$ (which is inferior to $0.69$ when using the signal vector).

*Table 10.* Network accuracy and confusion matrix for the **corrected** network trained on the Chess Pieces dataset. Rows correspond to ground-truth labels and colums to network predictions. Three classes are considered: **b**ishop/**k**night/**r**ook. The rows of the confusion matrix are normalized against ground-truth element count. Three datasets are also considered: **Clean** (without watermarks), **Poisoned** (with watermarks) and **Clean & Poisoned** which is the union of the previous two.

| *Dataset:* | **Clean** | | | **Poisoned** | | | **Clean & Poisoned** | | |
|---|---|---|---|---|---|---|---|---|---|
| *Accuracy:* | 0.93 | | | 0.69 | | | 0.81 | | |
| | **b** | **k** | **r** | **b** | **k** | **r** | **b** | **k** | **r** |
| **b** | 0.95 | 0.05 | 0.0 | 0.4 | 0.3 | 0.3 | 0.67 | 0.17 | 0.15 |
| **k** | 0.0 | 0.95 | 0.05 | 0.0 | 0.85 | 0.14 | 0.0 | 0.90 | 0.10 |
| **r** | 0.04 | 0.04 | 0.91 | 0.0 | 0.18 | 0.81 | 0.02 | 0.11 | 0.87 |

