# OpenReview forum: "Learning Encoding-Decoding Direction Pairs to Unveil Concepts of Influence in Deep Vision Networks"
_ICML.cc/2025/Conference — Submitted to ICML 2025_

### Official Review · Reviewer_eDaT · 2025-03-14

**Overall Recommendation:** 2

**Summary:**

This paper proposes a series of elements for boosting the unsupervised learning of encoding-decoding direction pairs. Specifically, the paper proposes a signal-distractor model to encode various concepts and distractor components, several regularization losses to optimize the upper bounds and enforce the sparsity, signal vectors to estimate the concept, and losses to align uncertainty regions between the network and the concept detector.

**Claims And Evidence:**

Some of the claims, such as the advantage of $L^{uur}$, $L^{cur}$, and other loss terms, are supported by the ablation studies. However, the paper lacks sufficient meaningful evidence to support its overall advantage over previous SOTAs. Details will be discussed in the following.

**Essential References Not Discussed:**

N/A

**Experimental Designs Or Analyses:**

Some of the experiments on the real dataset seem to be fine, but the experimental design on the synthetic datasets is not convincing. Also, what dataset is used in Table 3? Synthetic dataset or any of the two real datasets? Further, the description of the "top part of Table 2" and "lower section of Table 2" is quite confusing. I initially thought the "top" and "lower" part meant I = 500 or 450.

**Methods And Evaluation Criteria:**

The experiments on Places365 and MiT seem to be fine, but I don't think the "experiment on synthetic data" is significant. Running experiments on synthetic data with a fully controlled setting is a valid experiment setup, but the "synthetic data" in this paper is way too simple. I am not able to find more details on how the data are synthesized, as Section E only describes the training process. To my best understanding, some images (probably with very few, maybe two, pixels as W = 2 and H = 1) are processed by very simple two-layer networks. Although this may reveal some advantages of the proposed element over SOTAs, I don't think such an oversimplified experiment setup can lead to any meaningful conclusions. Authors may consider generating some actual synthetic images with controlled settings for the experiments and include both quantitative results and sample visualizations.

**Other Comments Or Suggestions:**

I strongly suggest that the authors polish this paper significantly for clarity. The author should provide a straightforward explanation for readers to understand the background, the shortage of literature, the motivations of each proposed element, and how each element can resolve the identified problem. The paper can also include simple examples or illustrations to help readers understand high-level background and contributions.

**Other Strengths And Weaknesses:**

In general, I find this paper extremely hard to follow and understand. There are quite a lot of grammar errors, and a lot of sentences are very confusing to me. I feel like this paper may have several plausible contributions (such as the uncertainty region alignment), but the poor presentation restricted me from fully appreciating them.

**Questions For Authors:**

Please refer to the weakness and other comments.

**Relation To Broader Scientific Literature:**

I believe the topic of this paper, especially the uncertainty region alignments and the improvement in interpretability, has the potential to benefit certain downstream tasks.

**Theoretical Claims:**

The paper should elaborate more on several equations, like eq. 4 and 5. I am not suggesting that they are wrong, but more straightforward discussions can help the reader to understand them.

---

> ### Author Rebuttal · Authors · 2025-03-30
>
> We thank the reviewer for the detailed review and thoughtful suggestions, which can be instrumental in improving our work. We greatly appreciate the time and effort the reviewer invested in providing constructive feedback. Below you may find some clarifications regarding your specific comments.
>
> **Experiment on synthetic data**: While we understand the reviewer's concerns for the simplicity of the example, we still think that this might be due to a misunderstanding. For this reason please let us briefly clarify the main design points of the experiment and also argue why this is valuable and informative. First, we need to clarify that a) the method is **post-hoc** and b) the experiment takes place in the **representation** space, and not in the domain of raw pixels. In this space, we make the assumption that concepts are encoded in distict directions. We use randomly generated matrices $\mathbf{S}$ and $\mathbf{D}$ (provided in Table 6) and Eq (3) to encode those concepts. For simplicity, we consider image **representations** comprised of two "pixels" (spatial elements in the representation space) with each pixel belonging to a different concept. In the real world this representation may correspond to an image comprised of an image patch depicting the concept "tree" and a second depicting the concept "person", etc. You may also think that the real world images have spatial dimensions like 32x32, and after spatial pooling operations the image representations are comprised of just two spatial elements. Subsequently, given the (manual) encoding of the concepts in the representation space, we train a short network (which in real-world may correspond to the top part of a larger ConvNet) to classify the different "images" based on their concept content. The experiment demonstrates that the proposed approach is able to correctly identify the concept clusters and correctly estimate the concept encoding directions (the synthetically generated matrix $\mathbf{S}$). This experiment is valuable because it verifies, to some extend, the efficacy of the proposed approach in estimating concept directions with **known ground-truth**, something that is otherwise particularly challenging with real data. The works that we extend and complement, (Kindermans, Pahde) also consider  working directly in the representation space, as we do. The example may seem simple, nonetheless, it is significantly more complex than examples presented in the prior works, where only a binary concept encoding model was considered.
>
> **Table clarifications**: Table 3 is referenced in Section 5.2 and thus refers to the experiment on Resnet18 trained on Places365 as mentioned in the first sentence of the section. Regarding the top/bottom parts of Table 2, the reviewer's understanding is correct. Although $I$ varies between the parts, the ablation study aims to highlight different aspects of the method.
>
> **Comparison with previous SOTAs**: The reviewer stated that the advantage of the proposed method compared to previous SOTAs may not be well justified from the experiments. We think that in the simple case of the synthetic experiment we demonstrated a toy-case where the proposed approach was the only method to successfully recover the ground truth, among 1 unsupervised (SAE) and 1 supervised approach (PCAV) while we also theoretically justified the ineffectiveness of other unsupervised approaches (NMF and PCA) in a theoretically-grounded way. Regarding real-world evaluation against unsupervised SOTAs, prior works typically resort to subjective evaluations regarding the interpretability of the learned directions, lacking a quantitative protocol. For the purposes of this rebuttal we took a best-effort quantitative approach, and learned concept directions using PCA and NMF for the last conv layer of Resnet18 trained on Places365, with $I=500$ for both methods. Subsequently we used Network Dissection to label the directions, as we do for our method, but with the classification threshold learned as suggested in NetDissect (computed as the top 0.005 quantile of activations in each direction). We finally calculated the interpretability metrics to compare with ours, which are provided below:
>
> |   Method    | $\mathcal{S}^1$ | $\mathcal{S}^2$ |
> | :---------: | :-------------: | :-------------: |
> |     PCA     |      13.96      |      6.03       |
> |     NMF     |      36.26      |      17.21      |
> | EDDP (ours) |    **57.34**    |    **38.36**    |
>
> We think this table together with Table 5 and Section 5.1, supports the strength of our method compared to prior unsupervised works in both real and synthetic data.

---

> > ### Comment · Reviewer_eDaT · 2025-04-05
> >
> > I appreciate the detailed rebuttals from the authors. However, my concern about the synthetic dataset is still outstanding. I understand that the samples are synthesized to represent the "visual features" rather than raw image pixels, but this still does not justify that the "synthesis feature dataset" is sophisticated enough to serve as a major dataset to evaluate the performance of the concept encoder/decoders. Yes, we can conceptually "image" the synthesized features as "visual features" pooled from larger feature maps, which are encoded from RGB images by a vision model. But, eventually, they are different. Visual features are much more sophisticated than synthetic features (as described in Sec. 5.1). Also, although Kindermans et al. and Pahde et al. also work on the toy feature space, they specifically mentioned the "simple" dataset as a toy dataset, and they mostly rely on the real images for showing the advantages compared to previous works (e.g., Kindermans et al. used ImageNet and Pahde et al. used ISIC2019, Bone Age, and FunnyBirds dataset). In the paper, the only valid comparison with previous works is Tab. 5, and the proposed method only achieves comparable performance with CBE on Office365 (as Tab. 3 compares the proposed method with the supervised method and underperforms the supervised method). On MiT, the proposed method outperforms CBE, which is a promising result. But overall speaking, the experiment setup is weak.

---

### Official Review · Reviewer_qm9x · 2025-03-15

**Overall Recommendation:** 3

**Summary:**

The paper is related to the work that tries to extract concepts to understand deep vision networks better, which are called concept-based explanations. The paper's contribution is an approach to learning the concept encoding-decoding direction jointly, in an unsupervised manner. The paper is strongly built on background work, which is presented clearly in the paper. The paper proposes different new losses to improve the interpretability of the influential concept detectors and to learn concept directions. Another interesting contribution is the uncertainty region alignment strategy to improve concept influence and interpretability, which is based on the hypothesis that uncertain network predictions are related to ambiguous concept information. An experimental study is provided, including experiments on synthetic data and experiments on deep image classifiers, as well as ablation studies on the different components of the proposed approach. The paper also contains some appendices that present more details on the technical contributions and the experimental studies.

**Claims And Evidence:**

+ The main claims of the paper are well-motivated, in relation to previous and related works, and experimental evidence is provided. Nevertheless, it's sometimes hard to distinguish between what comes from the core works on which the approach was built and what's new. This aspect could be improved simply by highlighting the main contributions of the approach in a different way from Figure 1.

+ The main original contribution of the paper is the uncertain region alignment approach and the associated losses. The paper proposes an ablation study on the experiments done on deep image classifiers. The study shows the benefits of the uncertainty region alignment on both interpretability metrics and influence metrics. Nevertheless, it could have been interesting to better study, both theoretically and experimentally the way the uncertain region alignment is done and the underlying hypothesis. This uncertainty region alignment is a very nice idea and it could be more deeply studied.

+ The second important contribution of the paper is the multi-concept signal distractor data model and the filter signal vector orthogonality loss. For instance, in the vein of the work of [Marconeto et al](https://arxiv.org/pdf/2309.07742) on the inductive bias link to the concept extraction part or the work of [ Vielhaben et al](https://openreview.net/pdf?id=KxBQPz7HKh).

**Essential References Not Discussed:**

+ I would find it interesting to link this paper to all the literature on mechanistic interpretability. See, for example [Bereska et al., 2024](https://arxiv.org/abs/2404.14082) for a sound synthesis and, in particular, to link the key elements of the paper to the hypotheses supported by this literature: superposition hypothesis, linear representation hypothesis, etc.

**Experimental Designs Or Analyses:**

The investigation of the proposed method is quite thorough.The experimental study of paper is very broad and relatively complete. The appendices also provide comparisons with other supervised and unsupervised concept detection approaches and details on the notion of influence. A final, very interesting part is the use of the approach for model correction.

**Methods And Evaluation Criteria:**

Methods and evaluation criteria are sound.

**Other Comments Or Suggestions:**

+ In order to have a better understanding of the direction learning process, described in Appendix D, I suggestion could be to do a schema. Globally, the full learning pipeline is difficult to follow in the paper.

**Other Strengths And Weaknesses:**

+ The paper is overall well written and experiments are discussed in detail.

**Questions For Authors:**

+ Deep state-of-the-art models include vision transformers. The proposed approach is tailored to CCN-based models. In particular, the approach is firmly built on the work of Doumanoglou et al and on the assumptions made on the structure of the feature space. How could the proposed approach be extended to other assumptions on the feature space and the network architecture?

+ The sparsity in the semantic space of concepts is related to the cluster count $I$ which is thus an hyperparameter of the method. How to correctly fix it ?

**Relation To Broader Scientific Literature:**

The proposed contribution can be seen as a nice integration and extensions of :
+ The works of (Doumanoglou et al, 2023, 2024) which is a core component of the proposed method. New contributions are the losses, in particular the losses coming from the uncertain region alignment idea that enable to improve interoperability and influence.
+ The works of (Kindermans et el, 2017) and (Pahde et al 2024) and the idea of signal distractor model extended with multiple concepts.

**Theoretical Claims:**

The paper doesn't contain theoretical claims but many technical aspects. The details given in the paper and the different appendices are very appreciated. In particular, they are mandatory for a good understanding of the paper and its contributions.

---

> ### Author Rebuttal · Authors · 2025-03-30
>
> We sincerely appreciate the careful review and insightful comments provided by the reviewer, which can greatly contribute to enhance our manuscript. The time and effort of the reviewer in providing constructive feedback are truly valued.
> - **Mechanistic Interpretability**: This work is related to **post-hoc** interpretability and mechanistic interpretability (MI) literature and especially concept-based explanations. The main assumption of this work is the linear representation hypothesis, i.e. assuming that concepts are encoded in the directions of the feature space. Whenever this holds, the proposed method has the potential to be a good candidate to identify the latent directions. While not certain, it seems that the linear representation hypothesis may hold for various architectures, including CNNs and ViTs (For instance, SAEs make a similar hypothesis for ViTs). In particular, in a soft sense, and at least for CNNs, we think that this hypothesis is somewhat more valid whenever the penultimate layer of the image classifier corresponds to a GAP layer (as it is for ResNets). While we demonstrated the efficacy of the approach in ResNets, our method would benefit from conclusions regarding Universality in MI, for instance if we finally may answer whether linear representation hypothesis is valid for all penultimate GAP layers, etc.  Our work is also related to the literature that tries to identify monosemantic features (concepts) from polysemantic neurons. While empirical evidence for the polysemanticity of neurons is broad, the superposition hypothesis assumes that this happens whenever the network wants to represent more features than neurons. However, this is not the only case when polysemnatic neurons emerge (Bereska et al). While our work is closely related to identifying monosemantic (concept) directions, we did not explore the case of more concepts than neurons, yet, this can be a subject for future work. Finally, our Uncertainty Region Alignment losses are also related to activation patching. Activation patching has been found to be an effective method to identify meaningful circuits in ViTs. The typical approach is to patch network activations with alternative values, such as zeros, mean activations, etc. Our Uncertainty Region Alignment losses rely on activation patching, yet, to our knowledge, in a novel approach, not explored before.
> - **Other architectures, such as ViTs**: The main assumption that we make about the feature space is that concepts are encoded in feature space directions. This assumption is more valid as we transition towards the top layers of the network (see also studies cited in the paper). Lately, the idea of superposition has been largely explored for ViTs, grounded on the same assumption that features (e.g., concepts) are encoded as directions in the latent space. While we did not explore the efficacy of our method in ViTs, we think that this can be a good subject for future work. Our experiments demonstrate that the proposed approach is effective when applied to ResNets, and ResNets and ViTs share some architectural similarities, such as skip connections. This fact seems encouraging to motivate the applicability of the proposed work to ViTs, but still, we are not certain for its success, unless we try.
> Please also consider our answer in relation to Mechanistic Interpretability.
> - **The choice for cluster count $I$**: Nice point (!) in which, unfortunately, we do not have a very good answer. Our intuition would favor $I$ such that the interpretability of the clustering is better. However, other approaches exist, such as preferring the clustering that linearly separates concepts better, thus favoring the structure of the feature space instead of the alignment of the clustering with human intuition. These might not be the only viewpoints though, but at least these are the two that we have come up with and ended in trying to choose $I$ in a way that favors interpretability. More specifically, we have never experimented with $I>D$. In ResNets, we've quantitatively found that the larger the $I$ the better the interpretability scores. However, the latter is also susceptible to the interpretability evaluation protocol. Although we made a best-effort approach, the interpretability of the clustering is difficult to be objectively quantified. For this reason some recent works have turned to subjective evaluation, often complemented with other metrics (such as sparsity) which altogether may provide an indication of whether the clustering is good. In particular, we share some of the experiences with: https://transformer-circuits.pub/2023/monosemantic-features, especially regarding about the subjective nature of interpretability and the fact that no matter how we approach the problem, we cannot find a very satisfying solution for the choice of $I$. For more details on the limitations of the quantitative approach that we took, please also refer to Appendix G.

---

### Official Review · Reviewer_wWMw · 2025-03-23

**Overall Recommendation:** 3

**Summary:**

The paper introduces a method for jointly learning concept “encoding” and “decoding” directions in an unsupervised manner. It uses a combination of interpretability-driven loss terms (e.g., sparsity, margin constraints), and alignment with the network’s uncertainty region. Experiments on synthetic data and Resnet classifiers show improved concept interpretability and influence compared to previous approaches. Overall, the idea is original with some concerns about the effect of different components in the design.

I will make my comments here and then will adjust my rating after the authors' response.

**Claims And Evidence:**

I am wondering what is the basis and main evidence for this claim in the paper? "we empirically prove that the uncertainty
region of the model is informative and can be used to effectively reveal meaningful and influential concepts that impact model predictions."
Is it Table 2? I might have misunderstood this part: Would you clarify how you can derive this conclusion?

**Essential References Not Discussed:**

None that I can comment on.

**Experimental Designs Or Analyses:**

Yes, I mentioned my concern regarding the ablation studies in previous sections.

**Methods And Evaluation Criteria:**

1. In [1], the authors have shown that different architectures have distinct activations on regions of an image (e.g. being sensitive to smooth regions etc.). This will likely be applied to the core idea of this research as well, where different architectures might have biases in their encoding-decoding concepts. Also, prior works cited in this paper have investigated more than one architecture to ensure the generalizability of their approaches. I recommend including other architectures such as the one used in prior works. Also, including ViTs or SwinT would strengthen the work.

2. In Table 2, the ablation study does not appear to cover all combinations of the newly introduced losses. Since each loss is motivated by a distinct objective (e.g., interpretability, alignment, orthogonality), it would be very informative to see a complete ablation matrix systematically isolating and combining each loss term, specifically seeing the performance when all 5 terms are present. Moreover, the current results look somewhat mixed: certain metrics improve in one setting but degrade in another, making it difficult to pinpoint how each loss individually contributes to the final performance.



[1] Feather, Jenelle, et al. "Discriminating image representations with principal distortions." ICLR 2025.

**Other Comments Or Suggestions:**

N/A

**Other Strengths And Weaknesses:**

Figure 1 can be edited to be more intuitive so that by looking at it, the readers can get a basic understanding of the input, output, the intended utility of each loss term, and also where the network is placed. (having a legend to describe the terminology)

Overall the paper is well-written!

**Questions For Authors:**

I am curious to know how the authors compare this work with attention in transformers. What would be the benefits of this approach?

**Relation To Broader Scientific Literature:**

A method toward explainable AI to understand image encoders.

**Theoretical Claims:**

N/A

---

> ### Author Rebuttal · Authors · 2025-03-30
>
> We want to thank the reviewer for her/his thoughtful review and valuable suggestions, which can help improve the clarity and quality of our manuscript. We truly appreciate the time and effort she/he dedicated to providing constructive feedback. Below you may find specific answers to some of the points mentioned in the review.
>
> - **About uncertainty region being informative**:
> Evidence comes from comparing $\mathcal{L}^{uur}$ with prior methods (UIBE, CBE) that lack Uncertainty Region Alignment, as shown in Table 5. It demonstrates that even without our other contributions, this loss improves direction discovery in terms of interpretability. Further, Table 2 (top, I=500, third/fourth row) shows additional gains across all metrics when shifting from Unconstrained to Constrained Uncertainty Region Alignment, which also estimates concept encoding directions.
>
> - **Ablation study**: Ablating each and every other combination of the individual loss terms results into a considerable number of different experiments, which for the purposes of this rebuttal we cannot afford due to limited resources. However, let us try to explain a bit more on the results of Table 2. First, please consider that $\mathcal{L}^{uur}$ and $\mathcal{L}^{cur}$ are mutually exclusive. We either use the first or the second, with the second being the only to take into account concept encoding directions. Furthermore, $\mathcal{L}^{fso}$ makes sense only in the context of $\mathcal{L}^{cur}$ since, as we implied, $\mathcal{L}^{uur}$ does not consider signal directions. The top part of Table 2 (I=500) aims to ablate the interpretabilty losses introduced in this work. It starts from $\mathcal{L}^{uur}$, without considering any of the interpretability losses (first row), and gradually adds $\mathcal{L}^{sb}$ and $\mathcal{L}^{eac}$ (the first 3 rows). Finally, it concludes by considering both interpretability losses in the context of using the second variant of Uncertainty Region Alignment loss $\mathcal{L}^{cur}$, together with $\mathcal{L}^{fso}$. Although a clear winning combination across all metrics is not evident, we consider the combination of $\mathcal{L}^{cur}$ and $\mathcal{L}^{fso}$ to be the theoretically more sound and at the same time the most competitive across all metrics, since it performs best in $\mathcal{S}^2$ and SCDP and second best in terms of $\mathcal{S}^1$ and SDC. Moreover in the second-best metrics, this combination remains competent to the other most performing combination (first row).
> The bottom part of the same table (I=450), aims to ablate Uncertainty Region Alignment losses. For this purpose, the utilization of the interpretability losses are taken for granted. The first row considers $\mathcal{L}^{uur}$ while the second and the third consider $\mathcal{L}^{cur}$ and its combination with $\mathcal{L}^{fso}$. Once again a clear winning combination is not evident, but still the last row corresponds to the theoretically more sound and in the practical aspect more competent combination across all metrics.
>
> - **Position this paper in relation to ViT attention**: Transformer attention is often used as a saliency map to highlight objects in pixel space, offering a local explanation of "where the model is looking." In contrast, global explanation methods like TCAV and RCAV address "what the model is looking for" and whether a concept influences a class prediction. Our method enables this by identifying concept directions in the latent space, (a requirement for TCAV/RCAV) facilitating global explanations beyond ViT attention's local scope.
>
> - **Experiments in more architectures**: We appreciate the comment of the reviewer for exploring the efficacy of our method to other architectures. Unfortunately, in the scope of this rebuttal we did not have the resources (nor the space) to conduct (and present) a complete analysis. Some preliminary results, in interpretability terms are provided below for the penultimate layer of EfficientNet_b0 trained on ImageNet (I=1280).
>
> |                       Method                        | $\mathcal{S}^1$ | $\mathcal{S}^2$ |
> | :-------------------------------------------------: | :-------------: | :-------------: |
> |                    Natural-basis                    |      37.62      |       7.2       |
> |             CBE /w $\mathcal{L}^{uur}$              |      55.05      |      22.95      |
> | EDDP $\mathcal{L}^{cur} + \mathcal{L}^{fso}$ (ours) |    **56.34**    |    **29.43**    |
>
> We think the method is promising and exploring applicability to other architectures could be an interesting future direction. We hope that this acknowledged limitation (of less exploration in other architectures) will not discourage the reviewer from supporting the acceptance of our paper, especially when considering the several other contributions that we make. Please also refer to the comment to reviewer qm9x for other architectures.

---

> > ### Comment · Reviewer_wWMw · 2025-04-07
> >
> > Thank you for your responses!
> >
> > I have read the rebuttal comments, and they have addressed most of my concerns. However, with the version at hand and given the modularity of the work, I believe (at least) thorough empirical evidence is required to fully support the claims. While the authors have provided empirical evidence for some components, the current level of validation keeps the merit of the work within an acceptable range. Therefore, I maintain my original rating of "Weak Accept."

---

### Decision · Program_Chairs · 2025-05-01

**Decision:**

Reject

**Comment:**

The paper proposes a method to learn unsupervised encoding-decoding directions in the feature space. In particular, the model learns a concept detector and a set of learnable concepts by decomposing the latent code into a combination of these concepts and their relevance to the prediction. The authors also argue and empirically show that the model’s uncertainty region is informative and can be used to improve concept alignment. Experiments on both synthetic and real-world datasets are provided to support the modeling decisions and validate the claims.

The paper generally received borderline scores, with the main criticisms focusing on: 1)	the fair but potentially limited experimental validation, 2) the fact that the contribution could be strengthened by including further analysis of the uncertainty part and considering additional architectures, and 3) the clarity and accessibility of the paper.

Reviewers appreciated the provided experiments, but noted that, as a primarily empirical paper, further experimental validation would have been beneficial. Similarly, the analysis of the uncertainty contribution could be more extensive, and the applicability of the method to additional architectures could be explored. Lastly, the accessibility of the paper could be improved by more clearly highlighting the key contributions and, for instance, including visual illustrations to help non-expert readers grasp the main ideas.

Overall, the paper constitutes a reasonable submission that proposes some interesting ideas, but it requires further adjustments before it can be considered ready for publication. I recommend that the authors take the feedback into account and update the paper accordingly.